# Global seasonal forecasts of marine heatwaves

Michael G. Jacox[1,2,4 ✉], Michael A. Alexander[2], Dillon Amaya[2], Emily Becker[3], Steven J. Bograd[1,4], Stephanie Brodie[1,4], Elliott L. Hazen[1,4], Mercedes Pozo Buil[1,4] & Desiree Tommasi[4,5]

Marine heatwaves (MHWs)—periods of exceptionally warm ocean temperature lasting weeks to years—are now widely recognized for their capacity to disrupt marine ecosystems[1–3]. The substantial ecological and socioeconomic impacts of these extreme events present significant challenges to marine resource managers[4–7], who would benefit from forewarning of MHWs to facilitate proactive decision-making[8–11]. However, despite extensive research into the physical drivers of MHWs[11,12], there has been no comprehensive global assessment of our ability to predict these events. Here we use a large multimodel ensemble of global climate forecasts[13,14] to develop and assess MHW forecasts that cover the world's oceans with lead times of up to a year. Using 30 years of retrospective forecasts, we show that the onset, intensity and duration of MHWs are often predictable, with skilful forecasts possible from 1 to 12 months in advance depending on region, season and the state of large-scale climate modes, such as the El Niño/Southern Oscillation. We discuss considerations for setting decision thresholds based on the probability that a MHW will occur, empowering stakeholders to take appropriate actions based on their risk profile. These results highlight the potential for operational MHW forecasts, analogous to forecasts of extreme weather phenomena, to promote climate resilience in global marine ecosystems.

Marine heatwaves (MHWs) affect marine ecosystems around the globe, with reported impacts including altered primary productivity, proliferation of harmful algal blooms, displacement of ocean habitats, changes to distributions and populations of marine species, increased human–wildlife conflict and fishery disasters[1,5,15–19]. Reliable forecasts of these climate extremes would help marine stakeholders to mitigate negative impacts and seize opportunities, thereby improving resilience through anticipatory decision-making[7]. A key step in that direction is the development of MHW predictions, which can be achieved by using operational global climate forecasts. Seasonal (that is, 1–12-month lead time) sea surface temperature (SST) forecasts are routinely used to predict the state of large-scale climate modes, such as the El Niño/Southern Oscillation (ENSO)[20,21], and for targeted applications, such as the NOAA Coral Reef Watch coral bleaching outlook[22]. Here we use these climate forecast systems to develop global predictions of MHWs and evaluate their skill over the past three decades. In doing so, we highlight the feasibility of predicting MHWs and provide a foundation for a much-needed operational MHW forecast system.

## MHW forecast skill

The MHW forecasts developed here show considerable skill on seasonal time scales (Fig. 1). Relative to random forecasts (Methods), the model MHW forecasts have significant skill nearly everywhere at shorter lead times (up to approximately 2 months), over large areas of the global ocean at lead times of 3–6 months and in some areas at even longer lead times (6–12 months). Forecast MHW probability is also related to MHW intensity, with low probabilities preceding non-MHW periods and higher probabilities preceding stronger MHWs (Extended Data Figs. 1 and 2). The degree of forecast skill is highly dependent on region, with the highest skill found in the tropics (particularly the eastern tropical Pacific) and portions of the extratropical Pacific (off the west coasts of North America and Patagonia, east of Australia). The most predictable regions are not necessarily those with the most intense MHWs; relatively poor MHW forecast skill occurs in much of the Southern Ocean and in Western Boundary Current regions, in which highly energetic and variable currents produce intense but relatively short-lived MHWs[12,19]. As forecast lead time increases, the global pattern of forecast skill is retained, but forecast skill degrades; at 10.5-month lead time significant skill is confined primarily to the Eastern Tropical Pacific and portions of the Indian Ocean, Indo-Pacific region and high-latitude Eastern Pacific. Similarly, whereas the patterns in skill described above generally hold throughout the year, there is a seasonal modulation of our ability to predict MHWs for specific regions (Extended Data Fig. 3). For example, in some regions 3.5-month lead forecasts are most skilful when initialized in boreal winter (for example, Northeast Pacific, Indian Ocean), whereas for other regions forecasts tend to be more skilful when initialized in boreal spring (for example, tropical Atlantic) or summer (for example, Coral Triangle, Eastern Tropical Pacific). As the forecasts are built on monthly data, their skill reflects an ability to predict longer-lived warming events (as

[1]NOAA Southwest Fisheries Science Center, Monterey, CA, USA. [2]NOAA Physical Sciences Laboratory, Boulder, CO, USA. [3]University of Miami, Miami, FL, USA. [4]University of California Santa Cruz, Santa Cruz, CA, USA. [5]NOAA Southwest Fisheries Science Center, La Jolla, CA, USA. ✉e-mail: michael.jacox@noaa.gov

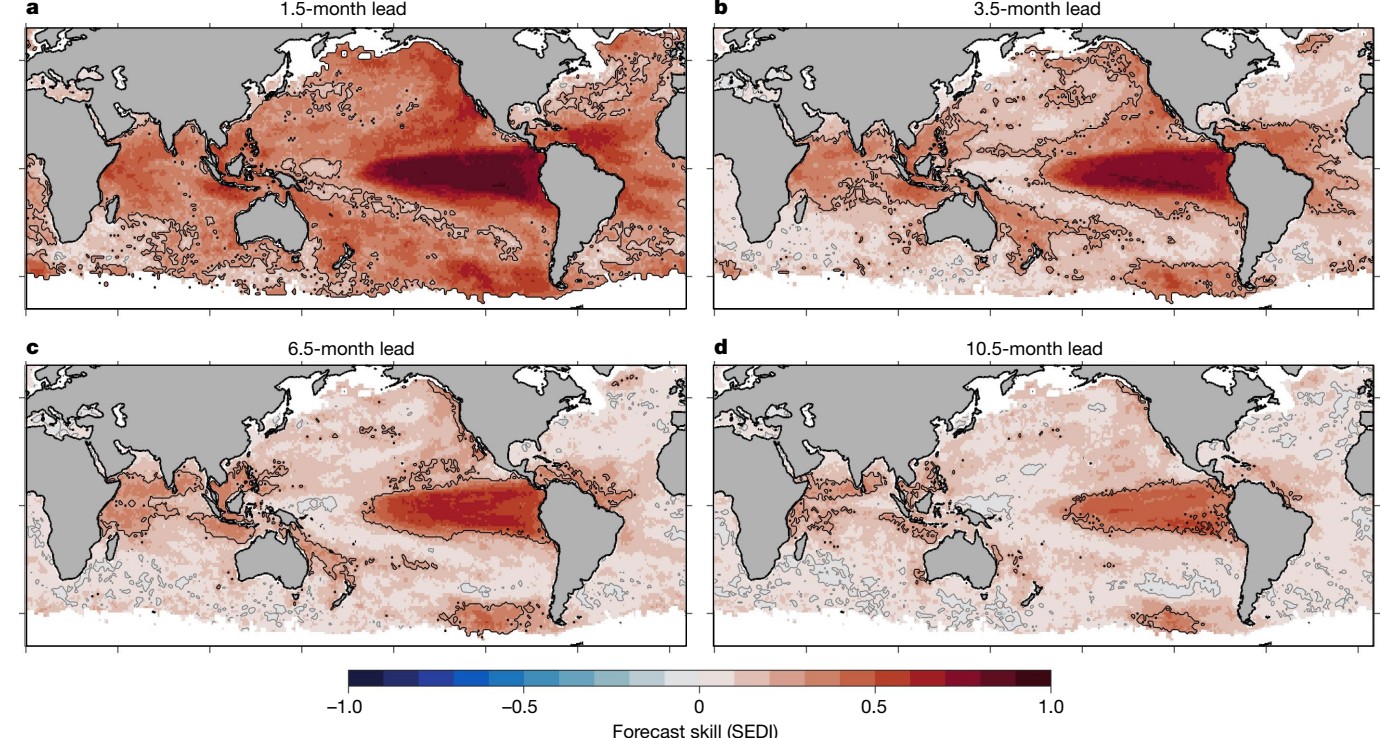

**a** 1.5-month lead **b** 3.5-month lead

**c** 6.5-month lead **d** 10.5-month lead

−1.0 −0.5 0 0.5 1.0

Forecast skill (SEDI)

**Fig. 1 | Skill of global MHW forecasts.** Maps indicate MHW forecast skill, as measured using the SEDI, for the 73-member ensemble of forecasts obtained from six global climate forecast systems for the period 1991–2020. SEDI scores range from −1 (no skill) to 1 (perfect skill). Scores above (below) zero, indicated by grey contours, indicate skill better (worse) than chance, and skill that is significantly better than random forecasts at the 95% confidence level is indicated by black contours. MHW forecasts were initialized every month, with lead times up to 11.5 months; a subset of lead times is shown here. **a**–**d**, 1.5 months (**a**), 3.5 months (**b**), 6.5 months (**c**) and 10.5 months (**d**). Areas with permanent or seasonal sea ice coverage are masked in white.

opposed to those lasting just days or weeks), although see the Methods (section 'Sensitivity to defining MHWs from daily versus monthly SST') for an exploration of forecasts using daily data.

From an end-user perspective, it is useful to quantify not just the overall forecast skill for MHWs (that is, whether there will be an active MHW in any given month; Figs. 1 and 2a), but also our ability to predict different characteristics of MHWs. Specifically, we focus on forecasting aspects of MHW evolution including their onset (that is, the month when a MHW first arises) and duration (how long a MHW persists once initiated). When averaged globally, forecast probabilities indicate elevated MHW likelihood (>10%) even 11.5 months before the observed start of MHWs (Fig. 2b). For shorter lead times (<3–4 months), forecast probabilities on average exceed 15% (that is, 50% higher than the base rate of occurrence) and increase rapidly as lead time decreases (Fig. 2b). However, there are substantial regional differences in the predictability of MHW onset. In regions with MHWs driven by rapid atmospheric or oceanic fluctuations, such as the Mediterranean Sea and western boundary currents like the Gulf Stream[11], skilful forecast lead times are often limited to two months or less (Fig. 2b) and even intense MHWs are unpredictable at longer lead times (Extended Data Fig. 2). By contrast, for regions in which MHWs result from predictable ocean evolution, such as the Eastern Tropical Pacific[21], highly elevated MHW probability (>20%, more than double the climatological probability) is forecast up to a year ahead of MHW onset. Regions influenced by atmospheric and oceanic teleconnections also show relatively high forecast skill; on average, the onset of MHWs in areas such as the Indo-Pacific region north of Australia, the California Current System and the northern Brazil Current are presaged by forecast MHW probabilities exceeding 20% approximately 3–6 months in advance (Fig. 2b and Extended Data Fig. 2).

MHW duration is highly variable across the world's oceans, with the mean length of events ranging from approximately 1 to 7 months globally (based on monthly SST data). We find that MHW forecasts

reproduce these spatial patterns well; over the ice-free regions of the ocean, there is a strong correlation between the mean durations of forecast and observed MHWs (Pearson correlation coefficient $r = 0.83$; Fig. 2c). However, not all regions show the same potential for accurately predicting the durations of different MHWs at a specific location. Temporal correlations between observed and predicted MHW duration tend to be highest in regions with the highest overall MHW forecast skill (Extended Data Fig. 4; compare with Fig. 1). Regions of higher skill also tend to have longer MHWs on average. Forecast skill (symmetrical extremal dependence index, SEDI) and mean MHW duration are strongly positively correlated (Pearson correlation coefficient $r = 0.74$), as regions with shorter duration MHWs tend to be less predictable (for example, Gulf Stream, Mediterranean Sea), whereas longer MHW duration is associated with greater predictability (for example, Eastern Pacific).

In the patterns of MHW forecast skill described above, there is a clear imprint of large-scale climate variability. In particular, the dominant signal in global maps of forecast skill (Fig. 1) is the unmistakable signature of ENSO, which is consistent with ENSO effects on seasonal SST predictability more generally[23–25]. Previous work has shown that ENSO is strongly tied to an increased or decreased frequency of MHW occurrence in many regions[12] and, although changes in the frequency of MHWs do not necessarily translate to changes in forecast skill (Methods and Extended Data Fig. 5), there is an ENSO-related modulation of MHW forecast skill. When ENSO is active at the time that forecasts are initialized (that is, during an El Niño or La Niña event), MHW forecast skill is enhanced in many regions (Fig. 3). Thus, the ENSO state at forecast initialization can be used for a priori assessment of whether a forecast is more or less likely to be skilful. The most pronounced forecast skill increases occur in the Indian and Eastern Pacific Oceans, and the globally averaged MHW forecast skill is closely linked to ENSO. The highest global skill in our 30-year record occurred during the extreme 1997–98 and 2015–16 El

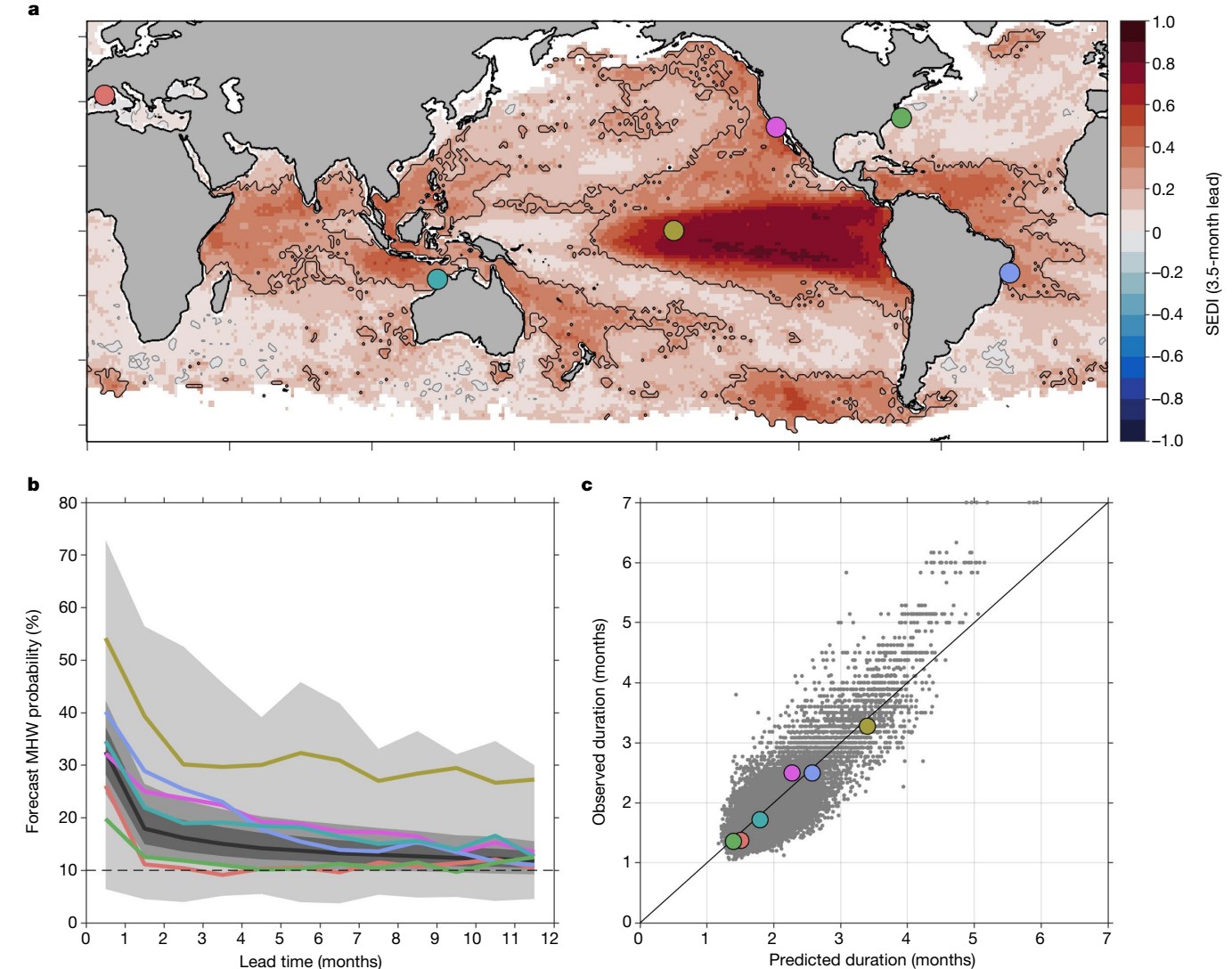

**Fig. 2 | Predicting the onset and persistence of MHWs. a**, SEDI for 3.5-month lead forecasts (as in Fig. 1). Example locations are indicated by coloured circles and are referred to in the text as Mediterranean Sea (red), Indo-Pacific (blue-green), Eastern Equatorial Pacific (gold), California Current System (pink), Gulf Stream (green) and Brazil Current (blue). **b**, Forecast MHW probability leading up to the initial appearance of observed MHWs. For each 1° × 1° grid cell, forecast probabilities for each lead time preceding the first month of observed MHWs are averaged across all events from 1991 to 2020. Coloured lines correspond to individual locations in **a**, whereas the grey line and shading indicate the global median and the 25th–75th, 10th–90th and 0–100th percentiles. For reference, a horizontal dashed line at 10% marks the

base rate of MHW occurrence; probabilities higher than 10% indicate that MHW forecasts correctly predict elevated MHW likelihood from 0.5 to 11.5 months in advance (for example, for 30% probability, forecasts are indicating that the likelihood of a MHW occurring has tripled). **c**, Comparison of observed and predicted mean MHW duration (that is, on average how long MHWs last once established at a given location). Each dot represents the mean duration of all events in a 1° × 1° grid cell, with coloured markers corresponding to locations in **a**. The strong correlation ($r = 0.83$) shows that the global spatial pattern of mean MHW duration is reproduced well by forecasts. For temporal correlations of observed and predicted MHW durations at individual locations, see Extended Data Fig. 4.

Niño events, and additional periods of elevated skill occurred during the 1991–92 and 2009–10 El Niño events and the 1998–2000 and 2010–11 La Niña events. There is debate about how ENSO events will change under increased greenhouse gas forcing, with some studies suggesting they may become more frequent or extreme in the future[26], whereas others point to limitations of global climate models in the tropics[27] and argue that the ENSO amplitude is more likely to decrease[28,29]. In any case, these studies should be extended to explore the potential impacts of ENSO changes on the predictability of MHWs and other extreme phenomena.

## MHW forecasts for ocean decision-making

Given the impacts of MHWs on ocean ecosystems, there is a need for operational MHW forecasts to help ocean users prepare for and adapt to

these events. In particular, skilful forecasts of MHWs would provide an early warning to resource managers and ocean stakeholders who could act to mitigate potential ecosystem impacts or capitalize on new opportunities. MHW forecasts could also portend changes in the availability of target and bycatch species to recreational and commercial fisheries, giving both fishing fleets and managers forewarning so as to maximize sustainable practices[11,30–32]. For example, proactive fishery closures may reduce both economic losses and ecological risk during events such as the 2014–16 MHW that led to increased baleen whale entanglements in the California Current System[5,33]. In other cases, MHW forecasts could inform the allocation of increased resources to monitor sensitive sites[34,35] or guide strategic planning to minimize aquaculture losses[36]. To support such proactive, climate-ready management approaches, forecast time scales must match those required for end-users to manage

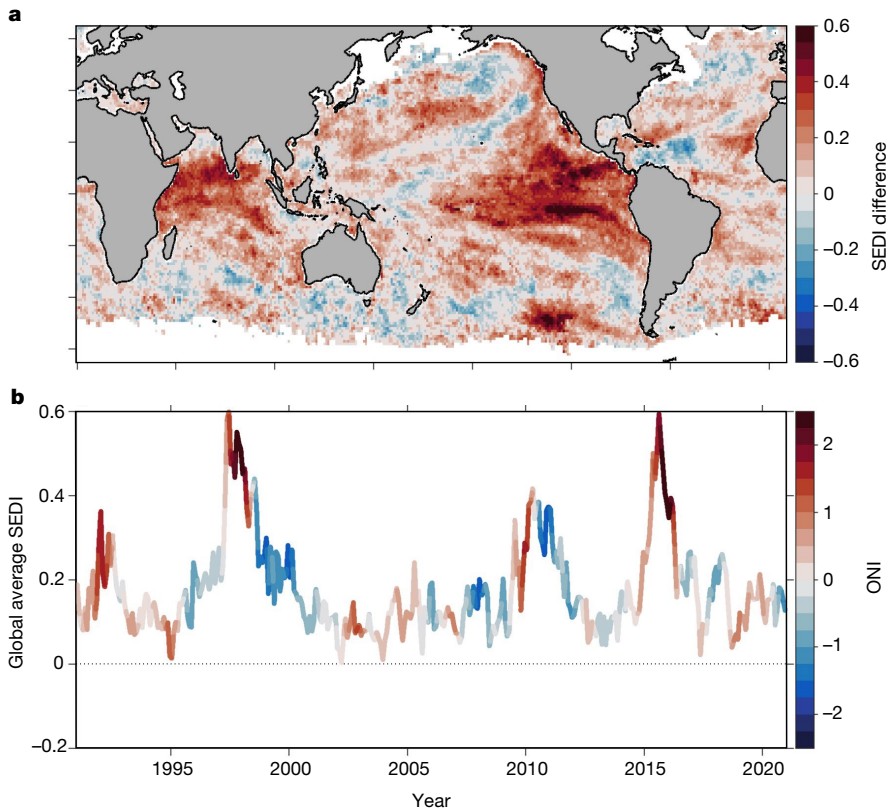

**Fig. 3 | Influence of ENSO on MHW forecast skill. a**, Difference in 3.5-month lead forecast skill (SEDI) between periods when ENSO is in an active state and when it is in a neutral state. Active states include both positive and negative phases, defined here as the upper and lower quartiles of the oceanic Niño index (ONI), respectively. **b**, Time series of globally averaged 3.5-month lead forecast skill, with ENSO state (as measured by the ONI) indicated by the colours. Although 3.5-month lead forecasts are shown here, the patterns of enhanced or suppressed skill also hold for other lead times.

climate risk and enact rapid adaptive responses[37,38]. Here the lead times of skilful MHW forecasts match the time scales of many marine resource management decisions[8], showcasing the potential for an operational MHW forecast system to be a highly effective decision support tool.

When utilizing probabilistic MHW forecasts, end-users will need to establish thresholds for the MHW probability at which decisions are triggered. The consequences of threshold choice are illustrated here for the Coral Triangle and the Galapagos Islands—two regions with coral reefs that are subject to bleaching or mortality during MHWs (Fig. 4). Enacting a lower threshold means that action is taken more often, which is a risk-averse approach that minimizes the rate of false negatives (failing to take action when a MHW occurs) but also leads to more false positives (taking action when a MHW does not occur). By contrast, higher thresholds limit action to more intense MHWs and a higher certainty of MHW occurrence (Fig. 4 and Extended Data Fig. 1) at the expense of an increased false negative rate. Individual users must balance the risk of inaction with the risk of unnecessary action—for example, trading off potentially adverse ecological impacts of unchanged ocean use during MHWs (false negatives) against economic consequences of excessive restrictions or excessive monitoring during non-MHWs (false positives). In this context, special consideration should be given to the handling of long-term SST trends in a forecast system, as the decision to retain or remove trends when defining MHWs will alter MHW frequency and consequently the statistics of forecast hits and misses (Methods and Extended Data Fig. 5).

## Operational MHW forecasts

The analysis here provides a template for, and demonstrates the feasibility of, an operational MHW forecast system to be used by ocean decision-makers. Because the MHW forecasts are built on the existing infrastructure of operational climate forecast systems, their transition from research to operations is relatively straightforward. In addition, the analyses performed here can be tailored to specific locations to provide site-specific decision support, including quantification of MHW forecast skill (Fig. 1, 2), its dependence on large-scale climate variability (Fig. 3) and appropriate decision thresholds (Fig. 4). In the future, our MHW forecasts could be expanded upon, with coupled climate forecasts from additional modelling centres and international collaborations (for example, from the Copernicus Climate Change Service; https://climate.copernicus.eu/seasonal-forecasts) as well as statistical forecasting methods such as linear inverse modelling[25,39] or machine learning techniques. In addition, whereas the monthly resolution of seasonal forecast output limits its application to the longer-lived MHWs (>1 month) that tend to be more predictable, forecasts of short-lived events may be useful and viable especially at short (for example, subseasonal) lead times. Regionally tailored MHW predictions can also be generated, either with statistical methods or by downscaling global forecasts, and may provide enhanced skill for specific areas. However, we anticipate that they would supplement, rather than replace, global forecasts. Ensuring global coverage facilitates equitable access to information about ocean extremes that may disproportionately affect regions and communities that lack the resources to develop regionally tailored MHW forecast systems[40]. Likewise, a global operational MHW forecast system can facilitate scientific collaboration to address the impacts of these extreme events on marine social–ecological systems. Given the pressing need for the forewarning of MHWs, the skilful predictions described here represent a key advance towards improved climate adaptation and resilience for marine-dependent communities around the globe.

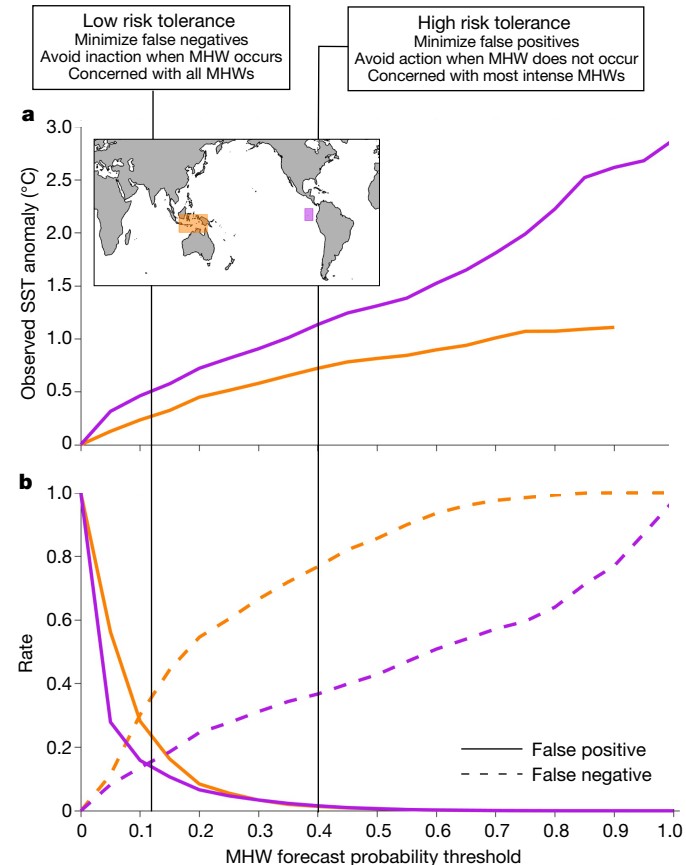

**Fig. 4 | Adjusting thresholds to support decision-making based on risk tolerance. a**, Observed MHW intensity (SST anomaly) shown as a function of MHW forecast probability threshold for 3.5-month lead forecasts in the Coral Triangle (orange) and Galapagos Islands (purple) regions. For a given threshold, SST anomalies are averaged over all times when the forecast probability was at or above that threshold. **b**, As in **a**, but for rates of false positives (solid lines) and false negatives (dashed lines). Note, **a** and **b** have the same *x* axis.

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

# Methods

## MHW observation

MHWs were identified based on v.2.1 of NOAA's Optimum Interpolation Sea Surface Temperature (OISST v.2.1)[41,42]. OISST v.2.1 was released in April 2020, and is identical to v.2.0 for data up until 2015 but includes significant quality improvements starting in 2016 (https://www.ncdc.noaa.gov/oisst/optimum-interpolation-sea-surface-temperature-oisst-v21). We obtained SST data at daily frequency and 0.25° horizontal resolution from NOAA's Physical Sciences Laboratory (https://psl.noaa.gov/data/gridded/data.noaa.oisst.v2.highres.html).

The bulk of our analysis was performed using monthly SST data for both observations and forecasts, but see the section 'Sensitivity to defining MHWs from daily versus monthly SST' for a discussion of the implications and practicality of using daily instead of monthly output. Daily 0.25° OISST data were averaged to monthly temporal resolution and 1° spatial resolution for consistency with the forecasts being evaluated (see the next section). MHWs were identified based on methods proposed in a previous study[43] and adapted for monthly data as described in ref. [19]. First, SST anomalies at each grid cell were computed by subtracting the 1991–2020 monthly climatology. MHW thresholds specific to each month of the year were then calculated as the 90th percentile of observed SST anomalies in a 3-month window (for example, for January MHWs, the 90th percentile of all December to February SST anomalies). SST anomalies were then converted to binary time series (MHW or no MHW) depending on whether they were above or below their respective thresholds.

## Global climate forecasts

Underlying the MHW forecasts described in this study are seasonal SST forecasts obtained from six global climate models contributing to the North American Multimodel Ensemble[13,14]. For each of the six models, an ensemble of forecasts is initialized each month, with the number of ensemble members and the forecast lead time varying between models (Extended Data Table 1). In addition to real time forecasts, a multidecadal set of reforecasts has been performed for each model. Reforecasts, also sometimes referred to as retrospective forecasts or hindcasts, are forecasts simulated for past periods using only information available at the time of forecast initialization (that is, ignoring information that has subsequently become available). The long historical suite of (re) forecasts is necessary to rigorously evaluate the skill and biases of the forecast systems. Here we obtained monthly averaged SST forecast output for 1991–2020, which is a period that is available from all six models, from the IRI/LDEO climate data library (https://iridl.ldeo.columbia.edu/SOURCES/.Models/.NMME/). Output from all models is served on a common grid with 1° resolution in longitude and latitude.

## MHW forecasts

To develop MHW forecasts based on the SST forecasts described above, a series of steps were performed for each model. First, the reforecast and forecast periods were concatenated to produce a single set of forecasts for analysis. For models that have more ensemble members in the real time forecasts than in the retrospective forecasts (Extended Data Table 1), we kept the same number of ensemble members as the retrospective forecasts to maintain consistency throughout the analysis period. Next, the model mean forecasts were calculated by averaging together the individual ensemble members of each model. The model mean forecasts were used to calculate model-specific monthly forecast climatologies for each initialization month and lead time, as is customary in climate forecast skill evaluation[44,45], and forecast anomalies were calculated for each individual ensemble member by subtracting the model mean climatology. Next, seasonally varying MHW thresholds for each model, lead time and initialization month were calculated with the same methodology described above for SST observations. Forecasts with SST anomalies at or above their respective thresholds were classified as MHWs, resulting in an ensemble of forecasts for binary outcomes (MHW or no MHW). The above steps were repeated for each of the six models, resulting in a multimodel ensemble of 73 members that was used to generate probabilistic monthly MHW forecasts. As forecasts are initialized at the beginning of the month, and we report monthly averages, lead times range from 0.5 months (for example, forecasts of January MHWs, initialized at the beginning of January) to 11.5 months (for example, forecasts of December MHWs, initialized at the beginning of January).

## Sensitivity to defining MHWs from daily versus monthly SST

In general, the time scale of predictable events increases with forecast lead time, such that one might look at daily output from weather-scale forecasts (for example, 1–2 weeks lead time) whereas monthly output is more appropriate for seasonal forecasts (up to a year). However, although the most impactful MHWs are overwhelmingly longer-lived events (>1 month)[46], there is also interest in more ephemeral warm extremes (lasting days to weeks) that may be missed in monthly averaged SST. To illustrate the influence of using daily rather than monthly SST forecasts for MHW prediction, we compare forecasts of MHWs identified based on daily and monthly output from CCSM4 for the locations highlighted in Fig. 2. For these locations, we obtained daily output of forecast SST for the entire 1991–2020 period from the CCSM4 model. We then repeated the analysis of observed and forecast MHWs using daily data with the definition described previously[43], which requires MHW thresholds to be exceeded for at least five days. Skill metrics for MHW forecasts generated from daily SST output were calculated using the same methods as those applied to monthly SST forecasts (see 'MHW forecast evaluation' below).

Relative to MHW forecasts defined from monthly data, forecasts at daily resolution show shorter mean MHW durations and often slightly lower skill, but no change in the reported patterns in MHW forecast skill (Extended Data Figs. 6 and 7). The consistency between the monthly and daily forecast skill is not surprising given that MHWs defined with daily data are still strongly driven by low frequency variability. However, it is important to note that even though seasonal forecasts can predict the enhanced or reduced likelihood of MHWs on daily time scales, this does not mean that one can predict the details of a specific short (for example, five-day) warming event months in advance. Rather, the skill in MHW forecasts provided at daily resolution is still reflective of predictable longer-lived SST anomalies, and forecast skill tends to be lower for shorter-lived events (Extended Data Fig. 6).

We also note that daily output is often not publicly available for seasonal forecasts (for example, NMME output is provided as monthly averages). Fortunately, we were able to get daily output from the CCSM4 model to conduct the comparison shown here, but at least in the near term a global MHW forecast system will necessarily be based on monthly output. The same may not be true for subseasonal forecasts (for example, 45 days or less), for which daily MHW forecasts would be more appropriate and daily model output would more likely be available.

## Accounting for warming trends

Owing to long-term warming trends in the world's oceans, the rate of MHW occurrence increases over time if fixed thresholds are used to identify them. This effect is prominent even over the relatively short 30-year period examined here, with MHW occurrence increasing two- to threefold if the warming trend is not accounted for (Extended Data Fig. 5). There has been debate in the literature about whether (or when) it is appropriate to retain or remove warming trends in MHW research[3,19,47]. Here we present results in the main text for MHWs calculated from detrended SST anomalies, but all analyses have been conducted using both methods. For the detrended analysis, we removed linear trends over the 1991–2020 period from the observed SST anomalies and the lead-time-dependent forecast SST anomalies at each grid cell. In the context of MHW forecasts, warming trends may be removed

or included depending on the user and the application, but it is important to understand the implications of how trends are handled. Some forecast skill metrics are sensitive to the rate of events, so if trends are retained (and MHW frequency increases over time), those skill metrics will also show trends that are unrelated to the actual capabilities of the model[48,49] (Extended Data Fig. 5). In the following section we expand on this point in the context of specific forecast skill metrics.

## MHW forecast evaluation

Our MHW forecast assessment follows common methods for evaluating climate and weather forecast skill, particularly for extreme events, which present challenges because of their relatively rare occurrence. For forecast verification, we first classify each ensemble member at each time step according to its position in the 2 × 2 contingency table: true positives (MHW is forecast and occurs), true negatives (no MHW is forecast and MHW does not occur), false positives (MHW is forecast but does not occur) and false negatives (no MHW is forecast but MHW occurs). From the contingency table we calculate two skill metrics, the forecast accuracy and the SEDI, described below. We also calculate the Brier skill score, which is derived from the MHW forecast probability (that is, the average of the binary forecasts from all ensemble members for a given forecast). Below, each of these metrics is described further. All three skill metrics show similar spatial patterns (Extended Data Fig. 8).

Of the many skill metrics proposed for forecasts of extreme events, SEDI[49] has several desirable qualities[50], including (1) it is non-degenerate, meaning that it does not trend towards a meaningless limit (for example, zero or infinity) as event rarity increases, (2) it is base-rate independent, meaning that it is not influenced by changes in the frequency of events, and (3) it is equitable, meaning its expected value is the same (zero) for random forecasts, regardless of what method is used to generate the random forecasts[51]. SEDI is calculated as

$$\text{SEDI} = \frac{\log F - \log H - \log\log(1 - F) + \log(1 - H)}{\log F + \log H + \log\log(1 - F) + \log(1 - H)},$$

where $H$ is the hit rate (ratio of true positives to total observed events) and $F$ is the false alarm rate (ratio of false positives to total observed non-events). The maximum SEDI score is one and scores above (below) zero indicate forecasts better (worse) than random chance.

For completeness, we also calculate two additional forecast skill metrics: the Brier skill score (BSS) and forecast accuracy. The Brier score is an estimate of the mean square error of the probabilistic forecast:

$$\text{BrS} = \frac{1}{N} \sum_{i=1}^{N} (f_i - o_i)^2$$

where $N$ is the total number of forecasts being evaluated, $f_i$ is the forecast probability computed from all ensemble members (that is, the fraction of forecasts predicting a MHW) for forecast $i$ and $o_i$ is the observed probability, which is either zero (no MHW) or one (MHW). The Brier skill score normalizes the Brier score relative to the skill of a reference forecast ($\text{BrS}_{\text{ref}}$):

$$\text{BSS} = 1 - \text{BrS}/\text{BrS}_{\text{ref}}.$$

Here the reference forecast is simply the climatological rate of MHW occurrence (that is, always predicting a 10% chance of a MHW occurring). The BSS ranges from one (perfect skill) to negative infinity (no skill); as for SEDI, scores above (below) zero indicate forecasts better (worse) than random chance.

Forecast accuracy is included as a common and easily understandable skill metric; it is simply the fraction of forecasts that are correct:

$$\text{forecast accuracy} = (\text{true positives} + \text{true negatives})/N.$$

For events that occur on average 10% of the time, the forecast accuracy for random forecasts is 0.82. Thus, MHW forecast accuracy above (below) 0.82 indicates skill better (worse) than random chance.

Significance of forecast skill metrics is quantified using a Monte Carlo simulation with block bootstrapping. Specifically, for a given grid cell we (1) calculate the MHW decorrelation time scale, $\tau$ (that is, the lag at which autocorrelation drops below 1/e), and then (2) randomly sample (with replacement) blocks of length $\tau$ from the observed MHW time series and concatenate them to create a forecast of length 360 months (the same as the model forecast verification period). This process is repeated to create 1,000 random forecasts, and forecast skill is calculated for each one. The 95% confidence intervals are then calculated from the skill values of the random forecasts, with significance defined as forecast skill exceeding the 97.5th percentile of the random forecast skill distribution.

When calculating time series of forecast skill (Fig. 3b and Extended Data Fig. 5), skill metrics are calculated over all grid cells at each time, rather than over all times at each grid cell. For example, the forecast accuracy for a given month in Extended Data Fig. 5b is the fraction of the ice-free global ocean for which the MHW state that month was correctly predicted. Temporal patterns in skill are similar between different metrics (Extended Data Fig. 5), with the exception that there is a base rate dependence in the forecast accuracy and in the individual components of the contingency table (true/false positives/negatives). That dependency is apparent during the strongest El Niño events (when SEDI and BSS increase but forecast accuracy declines), and also in the influence of long-term warming (Extended Data Fig. 5). If SST data are not detrended and consequently the rate of MHWs increases, then forecast accuracy declines, true and false positives increase, and true and false negatives decrease. These trends simply reflect changes in the frequency of events, whereas the performance of the forecast system (for example, as measured by SEDI) does not show a long-term trend (Extended Data Fig. 5). Thus, whether long-term temperature trends are removed or retained during MHW identification and forecasting, one must understand the implications for skill assessment.

## Data availability

NOAA High Resolution OISST v.2.1 data[41,42] were obtained from the NOAA/OAR/ESRL PSL, Boulder, Colorado, USA, at their website (https://www.esrl.noaa.gov/psd/). Global climate forecasts from the NMME[13,14] were obtained from the IRI/LDEO climate data library (https://iridl.ldeo.columbia.edu/SOURCES/.Models/.NMME/). The MHW forecasts described here can be accessed at the NOAA PSL MHWs page (https://psl.noaa.gov/marine-heatwaves/).

## Code availability

All analyses were performed using MATLAB. Codes can be accessed at https://github.com/mjacox/MHW_Forecasts.

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

**Acknowledgements** We thank N. Mantua for comments on an earlier version of the manuscript as well as A. Capotondi, C. Deser, J. Dias, K. Karnauskas, A. Phillips and J. Scott for discussions in the early stages of the study. This work was supported by funding from the NOAA Climate Program Office's Modeling, Analysis, Predictions and Projections program and the NOAA Fisheries Office of Science and Technology.

**Author contributions** M.G.J. conceived the study, performed the analysis and wrote the first draft of the manuscript. E.B. helped with downloading and processing the climate forecasts. All authors contributed to the design of the study, interpretation and presentation of results, and writing and revision of the manuscript.

**Competing interests** The authors declare no competing interests.

**Additional information**
**Correspondence and requests for materials** should be addressed to Michael G. Jacox.

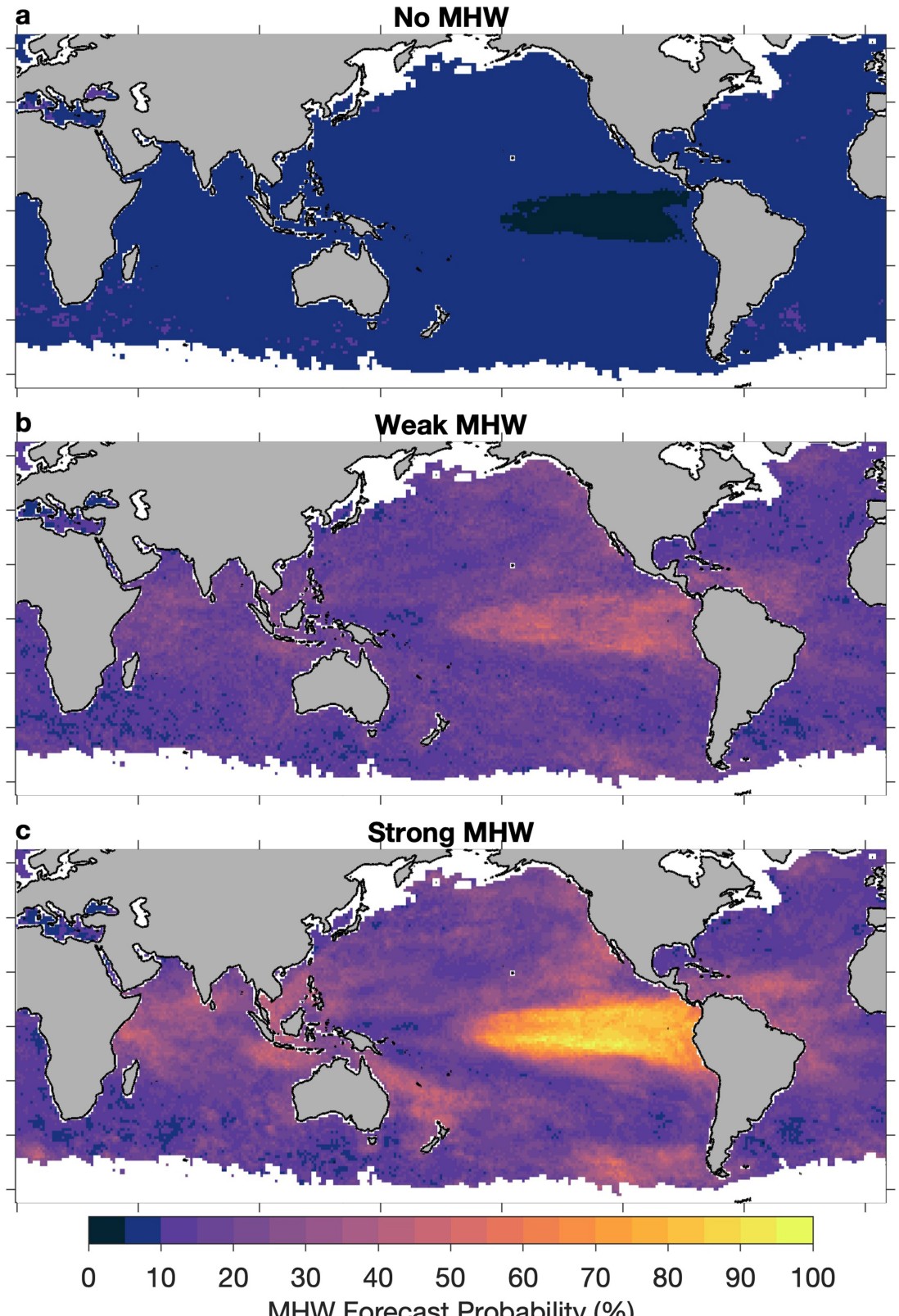

**Extended Data Fig. 1 | Forecast MHW probability varies with MHW intensity.** Maps show the mean 3.5-month lead forecast MHW probability associated with periods of **a**, no observed MHW (<90[th] percentile of SST anomalies) and observed MHWs that are **b**, "weak" (90[th]–95[th] percentile of SST anomalies) or **c**, "strong"(>95[th] percentile). Forecast probabilities higher (lower) than 10% indicate an elevated (reduced) likelihood of MHW occurrence. A positive relationship between MHW forecast probability and observed MHW strength is indicative of forecast skill.

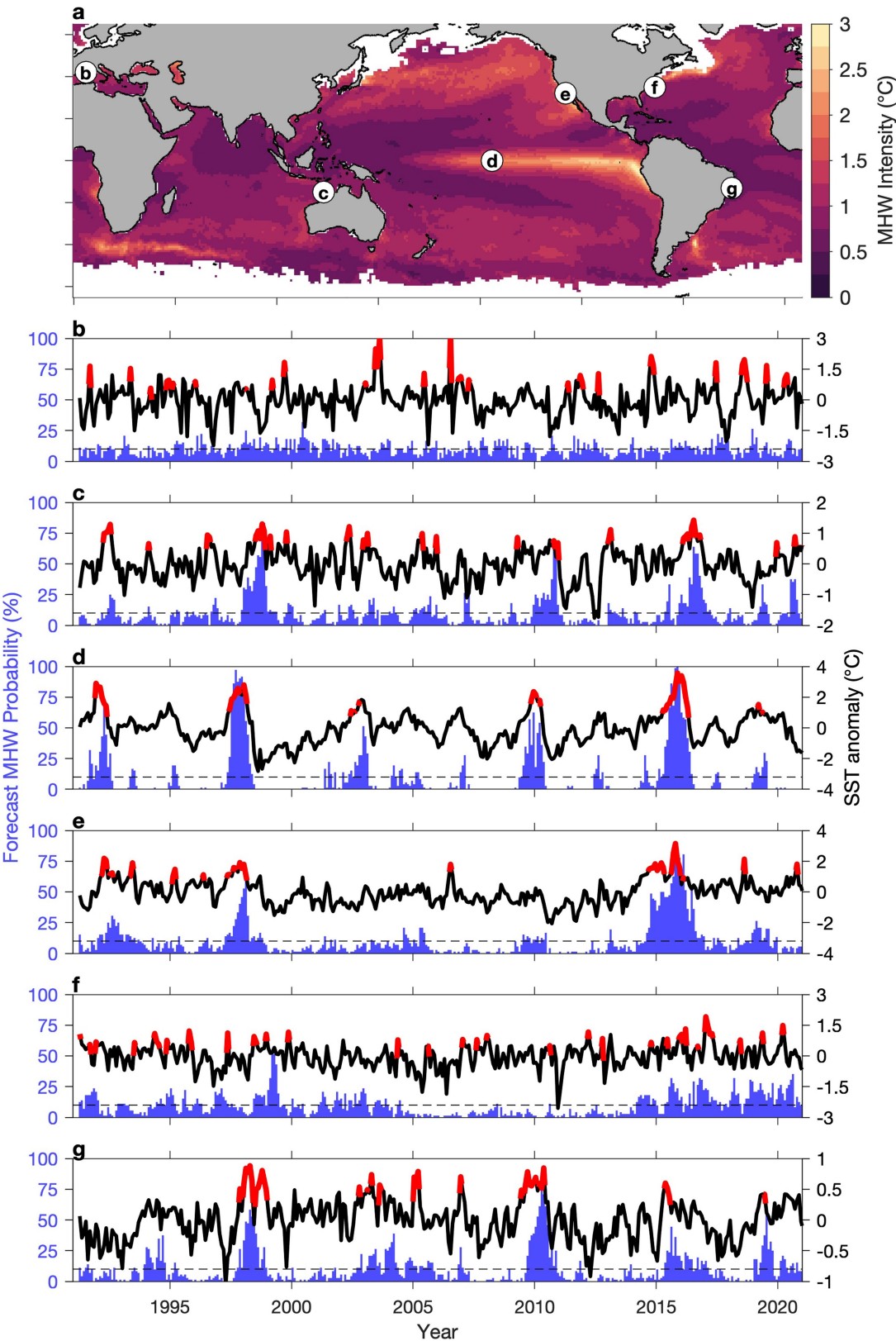

**Extended Data Fig. 2 | Observed and predicted MHWs for sample locations.** **a**, Mean observed MHW intensity (the average SST anomaly during MHWs), with markers corresponding to locations in Fig. 2. **b**–**g**, Time series of 3.5-month lead forecast MHW probability (blue bars) and observed SST anomalies (black, with MHWs indicated in red). Panel letters correspond to locations shown in **a**.

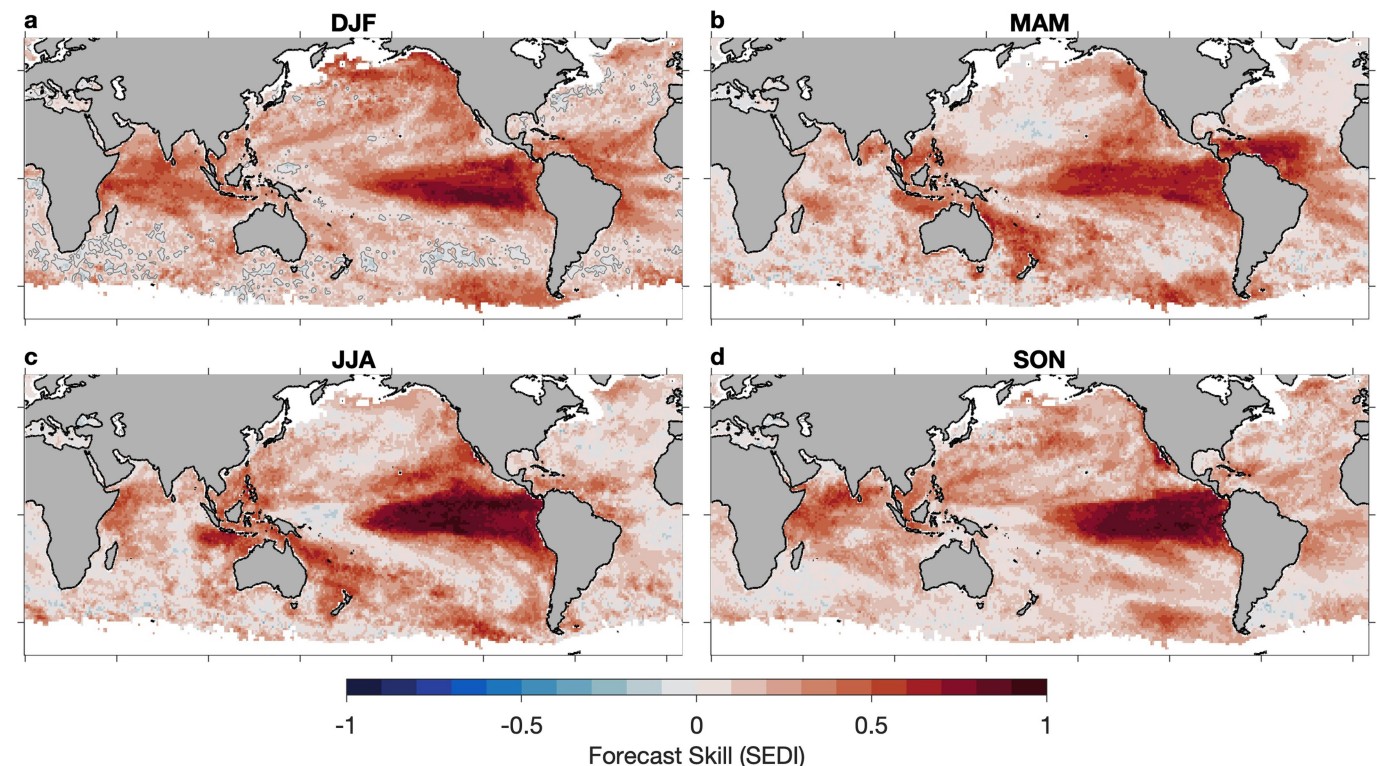

**Extended Data Fig. 3 | MHW forecast skill as a function of season.** Maps show 3.5-month lead forecast skill, as measured by the SEDI, for forecasts initialized in each season: **a**, December-February, **b**, March-May, **c**, June-August, **d**, September-November.

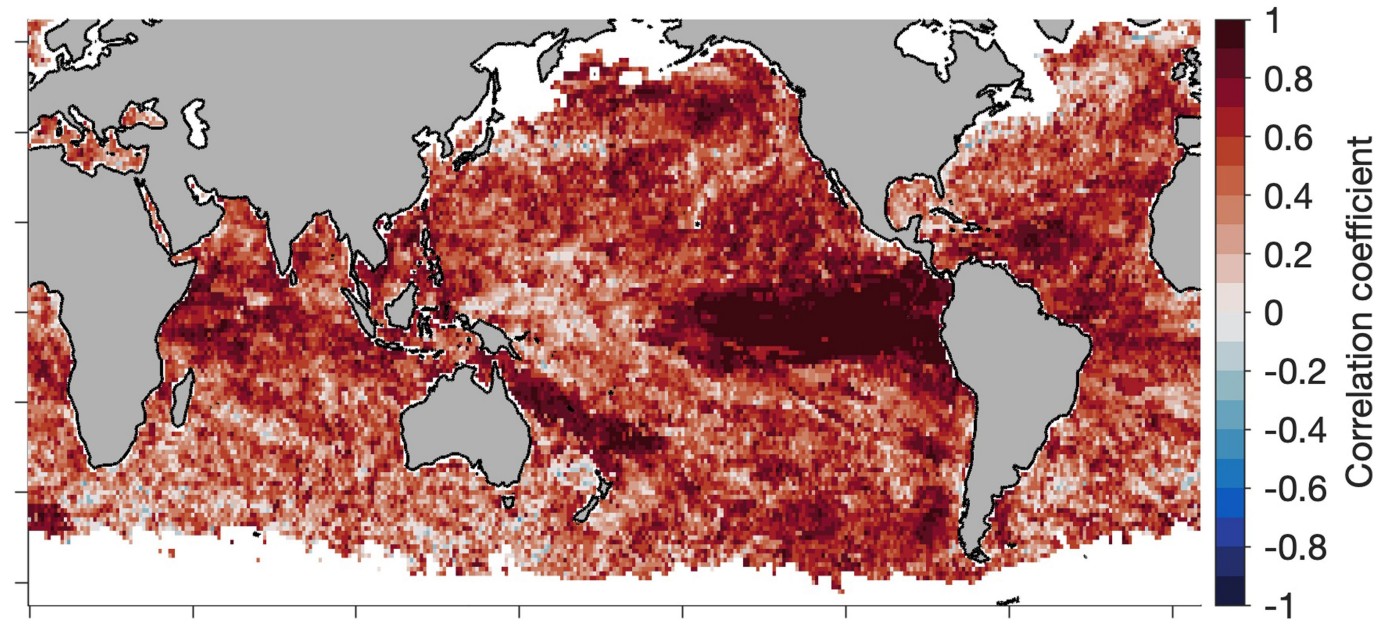

**Extended Data Fig. 4 | Comparison of observed and predicted MHW duration.** Maps show the correlation (Pearson correlation coefficient) between observed and predicted MHW duration at each location.

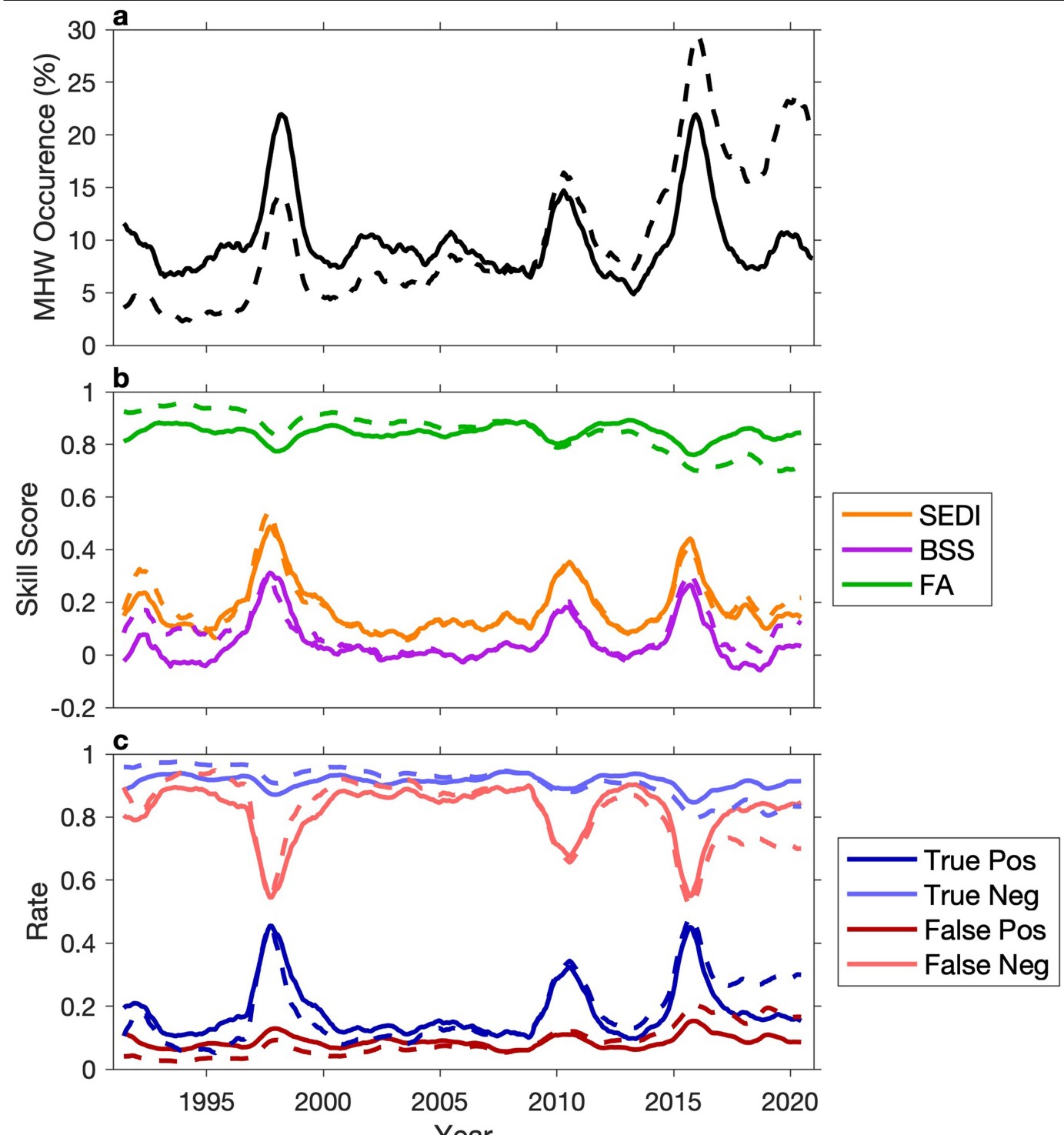

**Extended Data Fig. 5 | Temperature trends can influence MHW frequency and forecast skill metrics. a**, Time series show the global frequency of MHW occurrence (percentage of the ice-free global ocean in a MHW state at each monthly time step) calculated from SST anomalies with linear 1991–2020 trends removed (solid lines) and with trends retained (dashed lines). **b**, Time series of 3.5-month lead forecast skill metrics (Symmetrical Extremal Dependence Index, SEDI; Brier Skill Score, BSS; and Forecast Accuracy, FA). Skill metrics are calculated using globally aggregated forecasts each month (for example, forecast accuracy for a given month is the fraction of the ice-free global ocean for which the MHW state that month was corrected predicted). **c**, As in **b**, but for individual components of the 2x2 contingency table.

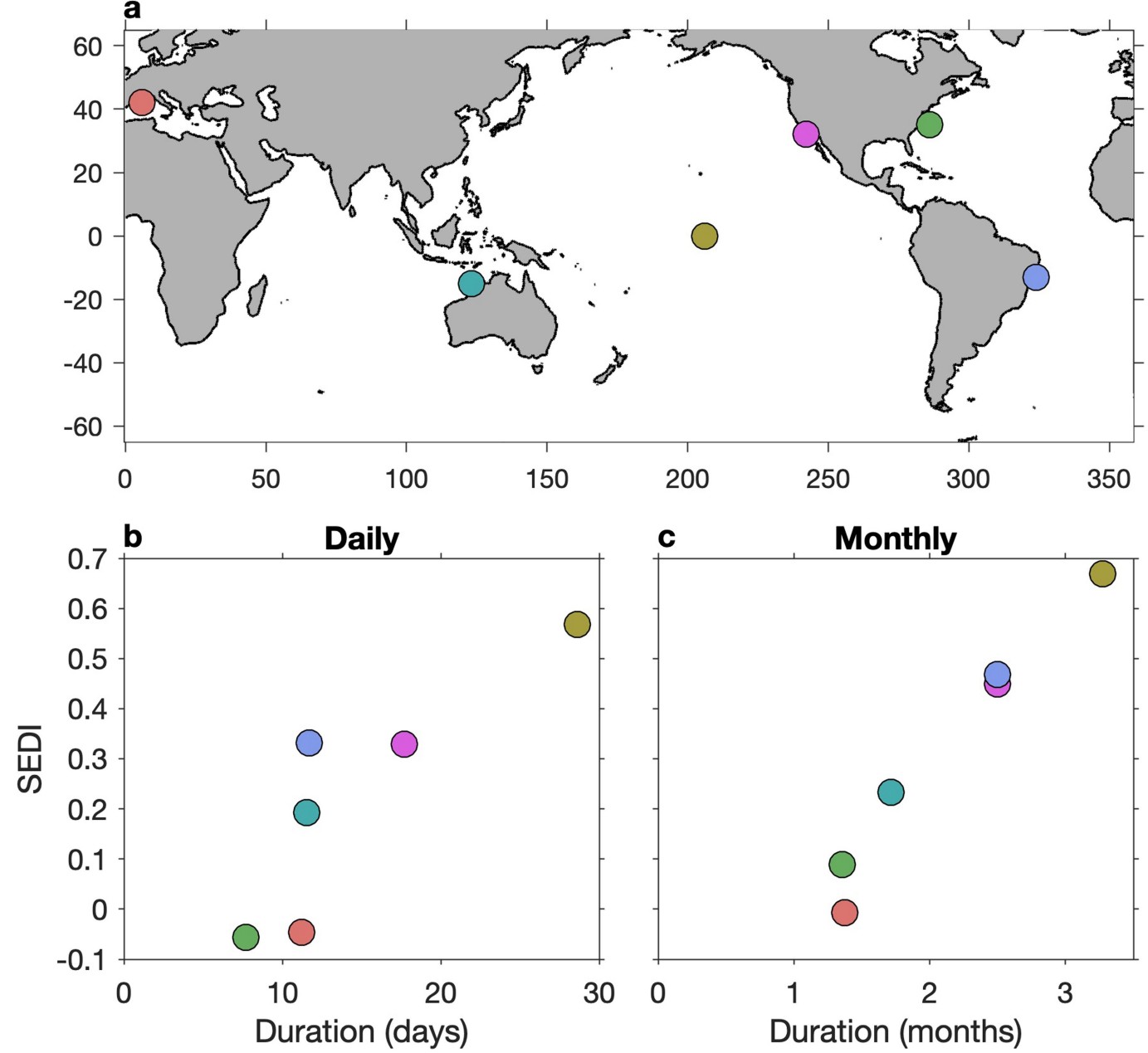

**Extended Data Fig. 6 | MHW forecast skill as a function of MHW duration for forecasts based on daily and monthly SST data.** For locations in **a** (which are the same as those in Fig. 2 and Extended Data Fig. 2), 3.5-month lead MHW forecast skill (SEDI) is shown as a function of mean MHW duration calculated from **b**, daily and **c**, monthly CCSM4 output.

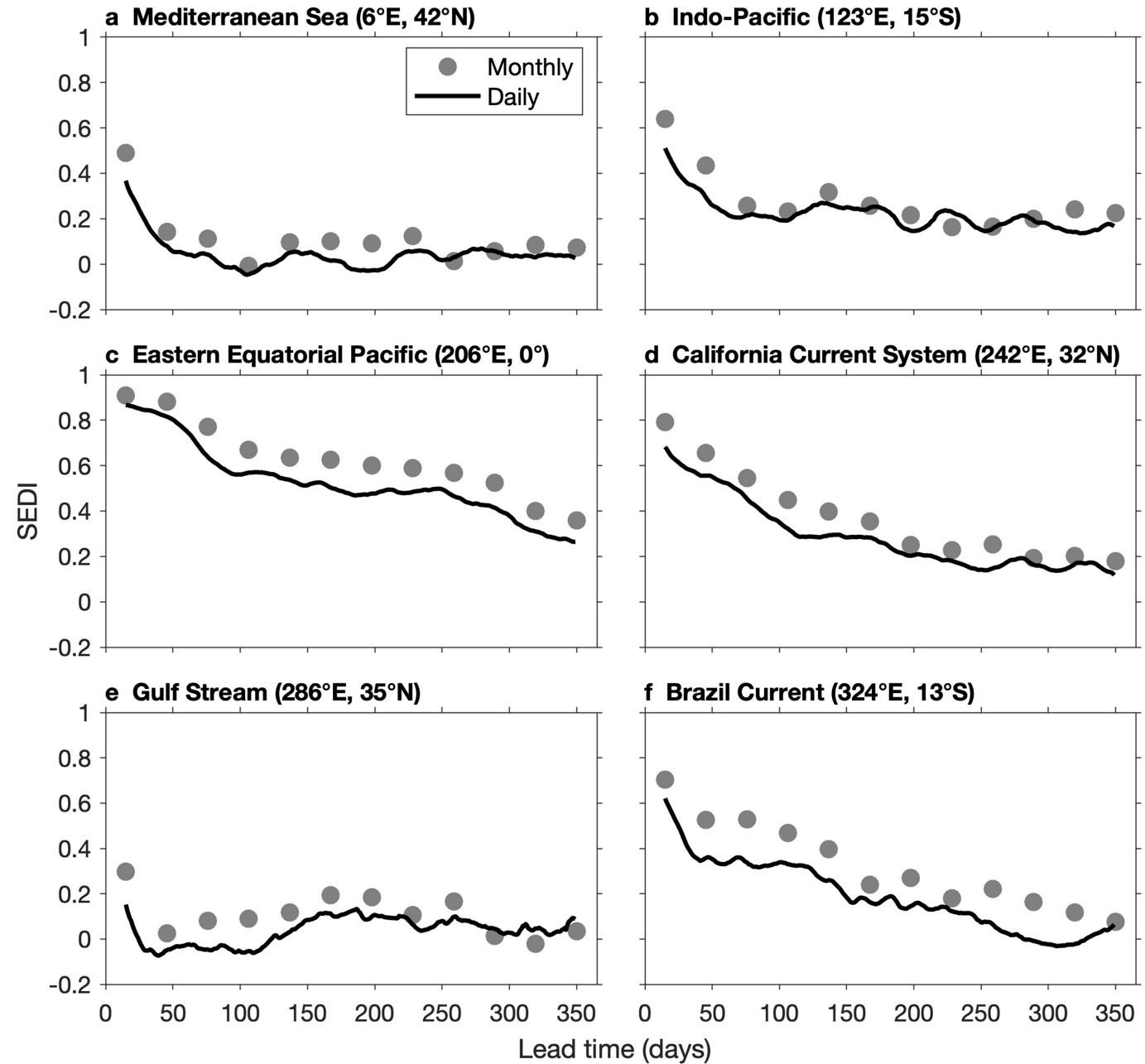

**Extended Data Fig. 7 | Comparison of lead time dependent MHW forecast skill for forecasts based on daily and monthly SST data. a–f,** For locations in Extended Data Fig. 6, forecast skill (SEDI) is shown as a function of lead time calculated from daily (lines) and monthly (circles) CCSM4 output. Daily skill is smoothed with a 30-day running mean for plotting.

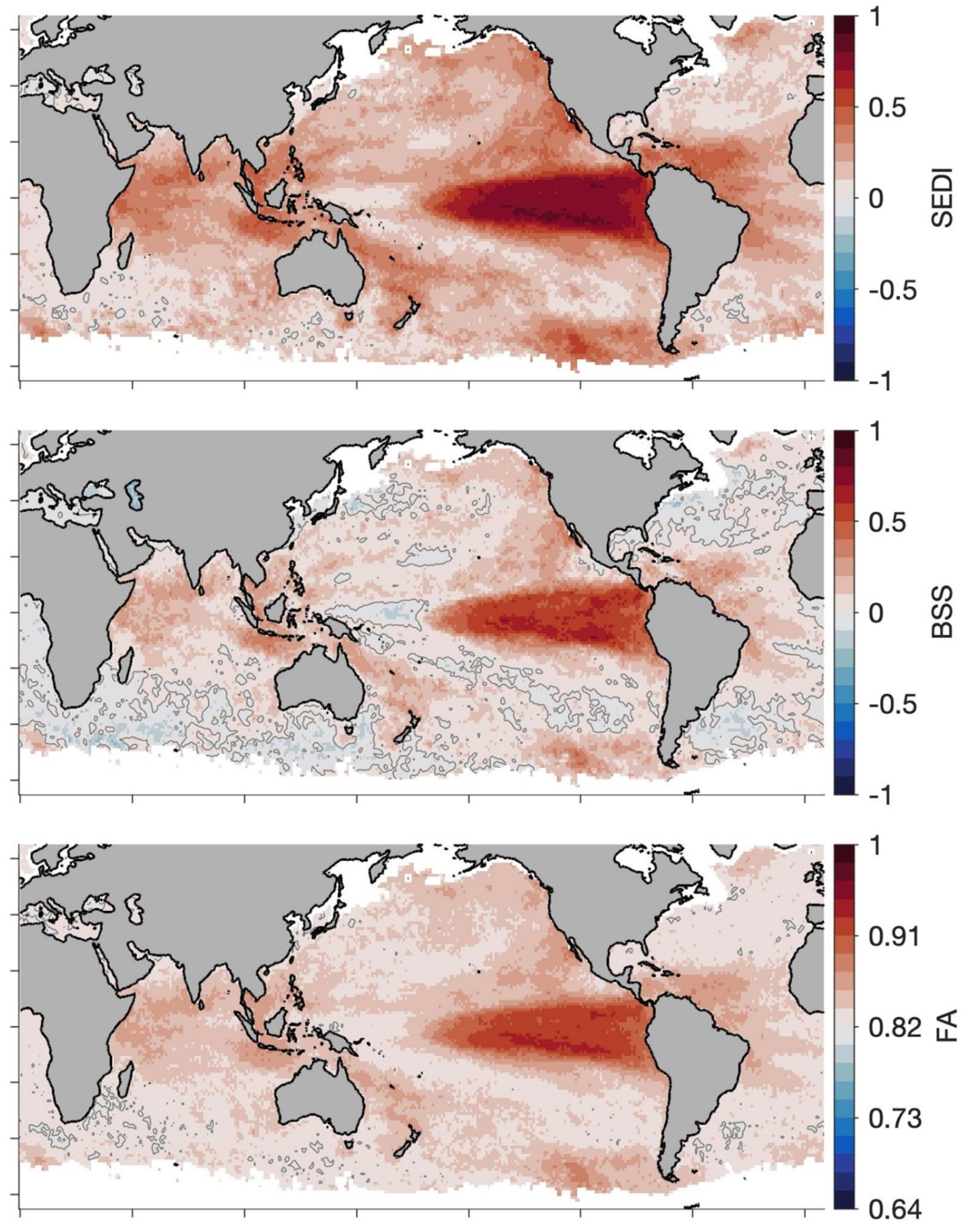

**Extended Data Fig. 8 | Comparison of MHW forecast skill metrics.** Maps show **a**, SEDI, **b**, Brier Skill Score (BSS), and **c**, forecast accuracy (FA) for 3.5-month lead MHW forecasts. Perfect forecasts would yield a score of one for all three metrics, while the skill expected from random forecasts is 0 for SEDI and BSS, and 0.82 for FA (indicated by gray contours).

**Extended Data Table 1 | Summary of NMME Forecasts**

| Model | Modeling Center | Ensemble Members | Lead Times (months) | Reference |
|---|---|---|---|---|
| CanCM4i | Environment and Climate Change Canada | 10 | 0.5-11.5 | Merryfield et al. (2013)[52] |
| GEM-NEMO | Environment and Climate Change Canada | 10 | 0.5-11.5 | Lin et al. (2020)[53] |
| SPEAR | NOAA Geophysical Fluid Dynamics Laboratory | 15 (retrospective) 30 (real time) | 0.5-11.5 | Delworth et al. (2020)[54] |
| GEOS-S2S | NASA Global Modeling and Assimilation Office | 4 (retrospective) 10 (real time) | 0.5-8.5 | Molod et al. (2020)[55] |
| CCSM4 | University of Miami | 10 | 0.5-11.5 | Infanti and Kirtman (2016)[56] |
| CFSv2 | NOAA National Centers for Environmental Prediction | 24 (28 in November) | 0.5-9.5 | Saha et al. (2014)[57] |

refs. [52–57]