## [Peer Review File · Nature]

Manuscript Title: Global seasonal forecasts of marine heatwaves

Reviewer Comments & Author Rebuttals

Reviewer Reports on the Initial Version:

Referees' comments:

Referee #1 (Remarks to the Author):

I was asked to evaluate the MS with focus on its use for predicting potential impact for management of ecosystems and fisheries. This, therefore, is the focus of my review.

The MS describes a method for predicting marine heat waves (MHW) based on seasonal forecasts from six established global climate models. The predictions surprised me by their apparent skills and capacity for forecasts; beyond what I had expected. Indications are that the forecasts may potentially be of use for resource management.

The approach is based on a method developed by Hobday et al, but is still original in how it is implemented and used. The MS acknowledges previous work, and is brief, clear and well-written.

As for the method, I was concerned (not working directly with how hindcasts are developed, but being an end-user) if there might be an issue with the hindcasts incorporating newer information. I understand (once getting through the Methods) that that is not the case for those used for this study, thus eliminating my only major concern.

As for the usefulness of MHW predictions for management of ecosystems and fisheries: this is important. I expect that the capabilities notably will be used in open ocean (tuna) fisheries for industry planning purposes at the quarter to year scale – which the method may be able to provide fairly reliable estimates for under the most important scenarios, i.e. where ENSO effects are at play. The economic consequences of that may be considerable. I also expect that the forecasts will be used for management, notably by RFMOs where the MHW predictions may impact boundaries and openings/closures.

For the modelling of future global ocean marine ecosystem and fish supply, (which I work with directly through FishMiP) it will be straightforward to incorporate predictions of MHW in our global model runs (certainly of those I'm involved with), and I expect it will impact the predictions we make about future seafood supply patterns and availability.

One aspect that is important for FishMiP long-term modelling, and which it might be useful to discuss in the MS, given how ENSO effects controls the globally averaged MHW forecast skills, is how reliable are long-term forecasts of future ENSOs? Last I read up on this, the expectation was that

ENSO events would be more frequent, (which we can work with), but not much more. So, an overview of this would be neat.

Villy Christensen

Referee #2 (Remarks to the Author):

The authors analyze the capability of current seasonal-to-annual climate prediction models to forecast the occurrence of anomalously warm ocean temperature conditions. They show that warm ocean surface temperatures can be skillfully predicted in most ocean regions on relatively short lead times of about 1-2 months. But skill decreases for longer lead times of 6 or more months, when skill is at first order restricted to those regions affected by ENSO. The authors further discuss the seasonal predictions as a potential tool for decision-making, and discuss how different probability thresholds may be relevant for different applications.

The study provides a comprehensive analyses of the predictability of warm ocean conditions based on existing climate prediction systems. In this context it primarily delivers a detailed evaluation of the prediction capability of these existing models, combined with a hypothetical discussion of how this may be useful for different decision-makers.

Overall the manuscript is well written and clearly presents the key concept and results. I have a few concerns about methodological details, which can have an effect on the exact meaning of the results, and have the potential to affect some of the conclusions of the results presented.

I believe that the definition used to identify marine heatwaves (MHWs) favors those events that are more predictable and excludes some less predictable ones, and thereby biases the results towards a more positive outcome. MHWs are typically considered as discrete but prolonged periods of anomalously warm ocean temperatures, and are characterized and can be detected by ocean temperature exceeding a certain threshold for a specified minimum period of time. By using monthly SST data to identify MHWs, the method applied here will mask out some heatwaves that (i) last less than a month or (ii) start peaking mid-month until around mid of the following month (and although the MHW lasts a month or longer it may not lift the calendar month values above the threshold when the other half of the month is cooler).

Using monthly SST data may therefore be suitable to detect some long-lasting MHWs (which are not necessarily the most intense events!) – and which typically would be driven by slowly varying modes of ocean variability that are more predictable. However, using monthly data may mask out events amplified by certain atmospheric drivers that typically act on shorter time scales, and that are less predictable. As outlined above, I argue that this choice of event definition has the potential to bias the detected MHWs towards those that are more predictable, and thereby may lead to overly positive skill results in particular in some extra-tropical regions. Such bias could be avoided when defining MHWs based on daily ocean temperature data. As the definition used here focuses the results on certain types of MHWs but likely excludes other events that should also be considered as MHW, I suggest a more accurate terminology of what is actually being analyzed here would be

“months with anomalously warm SSTs” (or similar) instead of MHW.

Another methodological concern is related to the significance estimates based on Monte-Carlo simulations (lines 296-301). If I understand correctly, this approach of randomly resampling binary MHW information assumes there is no auto-correlation in the occurrence of MHWs (or more accurately: monthly SSTs exceeding the 90th percentile). I have doubts whether this assumption is appropriate for regions where MHWs occur due to climate variations at lower frequency than monthly, e.g. when related to ENSO. In such cases it might be better to consider a block bootstrapping, where the length of blocks depends on the auto-correlation of MHW months in a certain region.

As another general comment, I was wondering how the prediction skill presented here for the occurrence of anomalously warm monthly average SST compares to the skill in predicting e.g. mean SST anomalies. Is the occurrence of monthly values in the upper decile simply a function of (predicted) monthly mean anomalies?

Other specific comments:

Line 1: the title should specify the time scale (seasonal to annual) of the forecasts being made

line 54: I'd say this is the climatology, or a climatological forecast, but not necessarily “random”?

Line 61: “extratropical Pacific” – it would be good to be more specific and mention explicitly the time scales and specific regions for which this is valid. In particular for lead times longer than 1.5 months it looks like there are larger areas in the extratropical Pacific where predictions are not skillful than where they are skillful?

Line 98-100: Not clear where the reader can see this relationship between MHW duration and predictability? Is this a general finding, or only valid for the 6 selected points?

Line 107-109: It may be good to specify in the text that this is the case (or is only shown at least) for 3.5 months lead time. Does it also hold for longer prediction lead times?

Line 162-163: Is there any evidence suggesting that downscaling may enhance prediction skill?

Line 303: the “temporal variability” of skill is not shown in Extended Data Fig. 5?

Figure 3b: Is the time series in panel b also for 3.5 months lead, as panel a? Please specify.

Extended Data Figure 4: It is not clear over which time period the skill values (e.g. in panel b) were calculated. I presume this is for moving windows shorter than the entire hindcast period? Please clarify.

Referee #3 (Remarks to the Author):

Review of Manuscript 2021-08-13716A authored by Jacox et al. (submitted to Nature)

Overarching Comments

The manuscript entitled “Global forecasts of marine heatwaves” by Michael Jacox et al., submitted to Nature, provides a global assessment of the predictability and forecast(hindcast) skill of marine heatwaves (MHWs) on 1- to 12-month time scales using a large multi-model ensemble over 30

years. This includes an analysis of the predictability and prediction skill seasonally and due to large-scale climate modes, in particular El Niño–Southern Oscillation (ENSO).

Overall, I found the paper to be interesting and the results mostly convincing. I expect this manuscript to be of broad interest to readers of Nature. I recommend the manuscript be revised in response to my comments below.

More Substantive Comments

LL59-64: Given that ENSO is most likely central to this pattern of MHW skill, is there anything that can be said here or elsewhere in the paper regarding the importance of ENSO to global scale predictability and the utility of MHW prediction skill?

LL64-66 re Fig 1: There is a distinct lack of forecast skill even at short lead times apparent along the South Pacific Convergence Zone – running from west equatorial Pacific east-southeast across the South Pacific. Can the authors comment on why this might be?

LL73-92 (and LL455-462), re Fig 2b: How is probabilistic forecasted onset accuracy/skill measured? Within a few days? Within a month? What is deemed to be a skilful probabilistic onset forecast?

LL94-102 (and LL462-465), re Fig 2c: The scatter plot does not connect back to the forecasts for each location. The authors should colour each of the predictions with the same colour used for region (i.e. outlined coloured circle). That would provide information that more clearly reflects the accuracy of the forecasts at each location. What are the individual correlation coefficients for each location?

Specific Comments

LL105-106: This is not surprising. This finding should be tied back to Fig. 3 of Holbrook et al. (2019), i.e. reference #12 in the manuscript.

L106: “be further” rather than “further be”

LL108-109: Does this specifically mean that when an El Niño or La Niña is declared, from that moment on the 3.5-month forecast begins? Or does it mean that the El Niño or La Niña are coincident with a MHW event that was forecast prior to the declaration of the ENSO event? Re Fig. 3b, does this represent when a 3.5-month forecast coincides with El Niño/La Niña? If so, I'm not sure that this is particularly impressive a finding. The authors should clarify.

LL111-113: It is not particularly surprising that larger ENSO events lead to higher MHW forecast skill.

LL115-117: I don't understand this statement. The authors should clarify what they mean here.

LL120-124: Do the authors really consider more skilful MHW forecasts would lead to a reduction in baleen whale entanglements? If so, how?

LL124-127: The authors might also cite reference #11 (Holbrook et al. 2020) again here as that paper focuses on and provides a comprehensive summary of the benefits to fisheries and aquaculture managers of forecasts specifically from MHWs.

LL135-149: This is interesting.

LL147-149: Is removing the trend for a 3.5-month forecast really needed?

LL474-481 re Fig. 4: This is a nice concept figure. One thing though is that it mixes observed SSTA (Fig 4a) and 3.5-month MHW forecast probability thresholds (Fig. 4b), which are somewhat different metrics. Presumably the decision-making by the user should take account of forecast skill, not only forecast probability, which underpins management choices regarding risk tolerances to false negatives and false positives. An interesting concept.

LL491-495 re Extended Data Figure 2: In Extended Data Fig. 2a, it would be good to see forecasts for just south of Tasmania in the apparently predictable region outlined in Fig 1a.

LL496-500 re Extended Data Figure 3: I find it interesting that for the Tasman Sea region, according to this analysis, MHWs are poorly forecast through the summer (DJF) when they tend to be largest and most impactful.

The reviewer is Neil Holbrook

Author Rebuttals to Initial Comments:

We are grateful to the three referees for their assessments of our manuscript and thoughtful input to guide its improvement. Based on your suggestions, we have performed additional analyses and revised the figures and text. Specific responses are included in italics below individual comments.

Referee #1 (Remarks to the Author):

I was asked to evaluate the MS with focus on its use for predicting potential impact for management of ecosystems and fisheries. This, therefore, is the focus of my review.

The MS describes a method for predicting marine heat waves (MHW) based on seasonal forecasts from six established global climate models. The predictions surprised me by their apparent skills and capacity for forecasts; beyond what I had expected. Indications are that the forecasts may potentially be of use for resource management.

The approach is based on a method developed by Hobday et al, but is still original in how it is implemented and used. The MS acknowledges previous work, and is brief, clear and well-written.

As for the method, I was concerned (not working directly with how hindcasts are developed, but being an end-user) if there might be an issue with the hindcasts incorporating newer information. I understand (once getting through the Methods) that that is not the case for those used for this study, thus eliminating my only major concern.

As for the usefulness of MHW predictions for management of ecosystems and fisheries: this is important. I expect that the capabilities notably will be used in open ocean (tuna) fisheries for industry planning purposes at the quarter to year scale – which the method may be able to provide fairly reliable estimates for under the most important scenarios, i.e. where ENSO effects are at play. The economic consequences of that may be considerable. I also expect that the forecasts will be used for management, notably by RFMOs where the MHW predictions may impact boundaries and openings/closures.

For the modelling of future global ocean marine ecosystem and fish supply, (which I work with directly through FishMiP) it will be straightforward to incorporate predictions of MHW in our global model runs (certainly of those I'm involved with), and I expect it will impact the predictions we make about future seafood supply patterns and availability.

Thank you for the support and additional insights on how these forecasts could be used.

One aspect that is important for FishMiP long-term modelling, and which it might be useful to discuss in the MS, given how ENSO effects controls the globally averaged MHW forecast skills, is how reliable are long-term forecasts of future ENSOs? Last I read up on this, the expectation was that ENSO events would be more frequent, (which we can work with), but not much more. So, an overview of this would be neat.

This is an interesting point and a good question. We have added a brief discussion of the topic at L116-121. Given ENSO's role in MHW predictability, any future changes to ENSO are likely to impact forecast skill. But the question of how ENSO may change in the future is an open one; some studies have suggested that ENSO events may become more frequent or extreme in the future (e.g., Cai et al. 2014), but others have pointed to limitations of global climate models in the tropics (Seager et al. 2019) and argued that ENSO's amplitude is more likely to decrease (Callahan et al. 2021, Wengel et al. 2021).

Villy Christensen

Referee #2 (Remarks to the Author):

The authors analyze the capability of current seasonal-to-annual climate prediction models to forecast the occurrence of anomalously warm ocean temperature conditions. They show that warm ocean surface temperatures can be skillfully predicted in most ocean regions on relatively short lead times of about 1-2 months. But skill decreases for longer lead times of 6 or more months, when skill is at first order restricted to those regions affected by ENSO. The authors further discuss the seasonal predictions as a potential tool for decision-making, and discuss how different probability thresholds may be relevant for different applications.

The study provides a comprehensive analyses of the predictability of warm ocean conditions based on existing climate prediction systems. In this context it primarily delivers a detailed evaluation of the prediction capability of these existing models, combined with a hypothetical discussion of how this may be useful for different decision-makers.

Overall the manuscript is well written and clearly presents the key concept and results. I have a few concerns about methodological details, which can have an effect on the exact meaning of the results, and have the potential to affect some of the conclusions of the results presented.

Thank you for the thorough review and constructive feedback. We have performed additional analyses and revised the text to address your methodological concerns.

I believe that the definition used to identify marine heatwaves (MHWs) favors those events that are more predictable and excludes some less predictable ones, and thereby biases the results towards a more positive outcome. MHWs are typically considered as discrete but prolonged periods of anomalously warm ocean temperatures, and are characterized and can be detected by ocean temperature exceeding a certain threshold for a specified minimum period of time. By using monthly SST data to identify MHWs, the method applied here will mask out some heatwaves that (i) last less than a month or (ii) start peaking mid-month until around mid of the following month (and although the MHW lasts a month or longer it may not lift the calendar month values above the threshold when the other half of the month is cooler).

Using monthly SST data may therefore be suitable to detect some long-lasting MHWs (which are not necessarily the most intense events!) – and which typically would be driven by slowly varying modes of ocean variability that are more predictable. However, using monthly data may mask out events amplified by certain atmospheric drivers that typically act on shorter time scales, and that are less predictable. As outlined above, I argue that this choice of event definition has the potential to bias the detected MHWs towards those that are more predictable, and thereby may lead to overly positive skill results in particular in some extra-tropical regions. Such bias could be avoided when defining MHWs based on daily ocean temperature data. As the definition used here focuses the results on certain types of MHWs but likely excludes other events that should also be considered as MHW, I suggest a more accurate terminology of what is actually being analyzed here would be “months with anomalously warm SSTs” (or similar) instead of MHW.

We appreciate that some of the MHW community is interested in very short-lived events (i.e., days to weeks). Based on your comments and the editor's, we computed forecast skill from daily and monthly forecast output for select locations previously highlighted in the manuscript. We added a new section "Daily vs. Monthly MHW Forecasts" in the methods and new Extended Data Figures 6 and 7. We also note in the main text (L63-66 and 166-169) that our results are focused on longer-lived MHWs and that daily resolution may be useful and viable especially for shorter lead times (e.g., from subseasonal forecasts).

While switching to daily forecasts does slightly lower skill in many cases, it does not change the reported patterns in MHW forecast skill. The consistency between them is not surprising given that even MHWs defined on daily are still strongly driven by low frequency variability.

Along with the additional analysis, we also felt it was important to discuss several points about the practicality and appropriate use of daily vs. monthly data for seasonal MHW forecasts:

- 1. In general, the timescale of events one might predict increases with the forecast lead time. While trying to predict daily variability would be appropriate for weather-scale forecasts (e.g., 1-2 weeks lead time), it is not well aligned with seasonal forecasts (up to a year). Even though MHW forecasts based on daily data can exhibit skill at long lead times, this does not mean that one can predict a very ephemeral (e.g., 5-day) warming event months in advance (which we think is exactly the reviewer's point). This distinction is an important one for end-users, i.e., even if you have seasonal MHW forecasts at daily resolution, their appropriate use is for predicting longer-lived events.*
- 2. Related to the point above, daily output is generally not available for seasonal forecasts. Fortunately, we were able to get daily output from the CCSM4 model to conduct the comparison in the manuscript. However, at least in the near term a global MHW forecast system will necessarily be based on monthly output, which again we argue is appropriate given that very short-lived events are not predictable far in advance.*

Last, as far as terminology, there is plenty of precedent for using monthly data for MHW analysis, as well as good arguments for why it may in fact be more appropriate than using daily data (e.g., atmosphere-ocean scaling, duration of most prominent events). Further discussion of this point and references for MHW studies using monthly data can be found in the methods of Jacox et al., Nature, 2020.

Another methodological concern is related to the significance estimates based on Monte-Carlo simulations (lines 296-301). If I understand correctly, this approach of randomly resampling binary MHW information assumes there is no auto-correlation in the occurrence of MHWs (or more accurately: monthly SSTs exceeding the 90th percentile). I have doubts whether this assumption is appropriate for regions where MHWs occur due to climate variations at lower frequency than monthly, e.g. when related to ENSO. In such cases it might be better to consider a block bootstrapping, where the length of blocks depends on the auto-correlation of MHW months in a certain region.

Good point, thank you. We redid the significance calculations using your suggested method, by block bootstrapping with block length equal to the local MHW decorrelation time scale. The

methods text has been revised to describe the new approach (L161-169 in the methods), which results in very small quantitative differences but no qualitative change.

As another general comment, I was wondering how the prediction skill presented here for the occurrence of anomalously warm monthly average SST compares to the skill in predicting e.g. mean SST anomalies. Is the occurrence of monthly values in the upper decile simply a function of (predicted) monthly mean anomalies?

Since MHWs are simply the warmest part of the SST distribution, all aspects of MHWs are closely related to the properties of underlying SST anomalies. For example, MHW intensity is a function of the variance in SST anomalies, MHW duration is a function of the autocorrelation of SST anomalies, etc. MHW forecast skill is similarly related to the SST anomaly forecast skill – for example, with the dominant role of ENSO (L103-105).

Other specific comments:

Line 1: the title should specify the time scale (seasonal to annual) of the forecasts being made

Done. The new title is “Seasonal marine heatwave forecasts for the global ocean”.

line 54: I’d say this is the climatology, or a climatological forecast, but not necessarily “random”?

The description of random forecasts here was incorrect. Apologies for the confusion - we have removed it and now refer to the methods section, where we describe the random forecast generation (which has been updated based on your comment above).

Line 61: “extratropical Pacific” – it would be good to be more specific and mention explicitly the time scales and specific regions for which this is valid. In particular for lead times longer than 1.5 months it looks like there are larger areas in the extratropical Pacific where predictions are not skillful than where they are skillful?

We revised the text to be more specific about which areas of the extratropical Pacific exhibit elevated skill and to specify how those regions change at longer lead times (L48-57).

Line 98-100: Not clear where the reader can see this relationship between MHW duration and predictability? Is this a general finding, or only valid for the 6 selected points?

This finding does hold in general. Across the ice-free ocean, mean MHW duration and forecast skill (SEDI) are correlated, with a Pearson correlation coefficient of 0.74. We have added this information to the text (L97-98).

Line 107-109: It may be good to specify in the text that this is the case (or is only shown at least) for 3.5 months lead time. Does it also hold for longer prediction lead times?

Yes, the same pattern of skill enhancement/suppression hold across other lead times. We have added this information to the caption of Fig. 3.

Line 162-163: Is there any evidence suggesting that downscaling may enhance prediction skill?

Good question, and it's still an open one. Anecdotally, we have found that dynamical downscaling provides little benefit for SST anomaly prediction in the California Current System, but that is only one region and downscaling may be more beneficial in others. We know there are people who think downscaling holds promise for improving forecasts, so we want to cover the possibility in the text.

Line 303: the “temporal variability” of skill is not shown in Extended Data Fig. 5?

We added a reference to Extended Data Fig. 4, which shows the temporal variability, and expanded the text describing the temporal variability of different indices.

Figure 3b: Is the time series in panel b also for 3.5 months lead, as panel a? Please specify.

Yes, panel b is also for 3.5-month lead. We have added this information to the caption.

Extended Data Figure 4: It is not clear over which time period the skill values (e.g. in panel b) were calculated. I presume this is for moving windows shorter than the entire hindcast period? Please clarify.

These are monthly time series of forecast skill; at each month skill is calculated by aggregating forecasts for the entire globe. e.g., forecast accuracy for a given month is the fraction of the ice-free global ocean for which the MHW state was correctly predicted. We have edited the caption to clarify this point.

Referee #3 (Remarks to the Author):

Review of Manuscript 2021-08-13716A authored by Jacox et al. (submitted to Nature)

Overarching Comments

The manuscript entitled “Global forecasts of marine heatwaves” by Michael Jacox et al., submitted to Nature, provides a global assessment of the predictability and forecast(hindcast) skill of marine heatwaves (MHWs) on 1- to 12-month time scales using a large multi-model ensemble over 30 years. This includes an analysis of the predictability and prediction skill seasonally and due to large-scale climate modes, in particular El Niño–Southern Oscillation (ENSO). Overall, I found the paper to be interesting and the results mostly convincing. I expect this manuscript to be of broad interest to readers of Nature. I recommend the manuscript be revised in response to my comments below.

Thank you for the positive feedback and helpful comments.

More Substantive Comments

LL59-64: Given that ENSO is most likely central to this pattern of MHW skill, is there anything that can be said here or elsewhere in the paper regarding the importance of ENSO to global scale predictability and the utility of MHW prediction skill?

Absolutely. The importance of ENSO to global scale predictability and MHW forecast skill is the focus of Figure 3 and the text at L102-121.

LL64-66 re Fig 1: There is a distinct lack of forecast skill even at short lead times apparent along the South Pacific Convergence Zone – running from west equatorial Pacific east-southeast across the South Pacific. Can the authors comment on why this might be?

Agreed, that is a striking feature. We don't know for sure, but can speculate with a couple of ideas. One is that the SPCZ is along a nodal line of the dominant ENSO pattern, which minimizes the ENSO-related forecast skill in the region (Newman and Sardeshmukh, 2017). Second, climate models have trouble reproducing the position of the SPCZ (Brown et al. 2020), and that bias could influence the forecast skill. Third, SST anomalies in the SPCZ are likely driven strongly by variable winds, precipitation, and clouds that are hard to predict. Given that these ideas are speculative, and that we can't give similar attention to each individual region, we prefer not to dwell on the SPCZ discussion in the text.

LL73-92 (and LL455-462), re Fig 2b: How is probabilistic forecasted onset accuracy/skill measured? Within a few days? Within a month? What is deemed to be a skilful probabilistic onset forecast?

Here, the measure of forecast accuracy for MHW onset is the forecast probability for the observed start (i.e., the first month) of MHWs. We edited the text and the Figure 2 caption to clarify. The point of this metric is to show the ability of forecasts to predict a MHW that isn't in place at initialization (as opposed to predicting the continuation of an existing MHW). Any forecast probability above 10% is a correct indication of elevated MHW likelihood, suggesting some level of skill. But in a practical sense, the threshold required for adequate certainty about a MHW forming would be determined by a decision maker for a specific region/application (e.g., Fig. 4).

LL94-102 (and LL462-465), re Fig 2c: The scatter plot does not connect back to the forecasts for each location. The authors should colour each of the predictions with the same colour used for region (i.e. outlined coloured circle). That would provide information that more clearly reflects the accuracy of the forecasts at each location. What are the individual correlation coefficients for each location?

To clarify, this figure is showing the mean observed and forecast durations, averaged across all MHWs at each grid cell. So, there is only one dot per location. You make a good point that this plot doesn't show the ability to forecast the durations of different events at a given location. To thoroughly show that information without making this figure too noisy, we have added Extended Data Figure. 4, which shows the correlation of observed and forecast MHW durations for all global locations. We also clarified the text with reference to the new figure at L90-96.

Specific Comments

LL105-106: This is not surprising. This finding should be tied back to Fig. 3 of Holbrook et al. (2019), i.e. reference #12 in the manuscript.

Based on this comment and the one at L115-117, we made revisions throughout this paragraph (L102-116) to clarify, including reference to Holbrook et al. (2019). While it may not be surprising that ENSO influences MHW forecast skill, it is not a necessary consequence of the changes in MHW occurrence shown in Holbrook et al. Figure 3. For example, if one leaves in the long-term warming trend, the frequency of MHWs increases but forecast skill doesn't (Extended Data Fig. 4). This point is also what we were trying to convey at L115-117 of the original manuscript (which did not come across clearly and has been revised).

L106: “be further” rather than “further be”

Changed

LL108-109: Does this specifically mean that when an El Niño or La Niña is declared, from that moment on the 3.5-month forecast begins? Or does it mean that the El Niño or La Niña are coincident with a MHW event that was forecast prior to the declaration of the ENSO event? Re Fig. 3b, does this represent when a 3.5-month forecast coincides with El Niño/La Niña? If so, I'm not sure that this is particularly impressive a finding. The authors should clarify.

We agree this wasn't clear. The modulation of forecast skill is shown for when El Niño or La Niña conditions are in place at the time the forecast is initialized. Based on this result we can say something about the likely skill of forecasts based on conditions when they are initialized, which is important for understanding and communicating conditional forecast skill (i.e., a priori knowledge of whether a forecast is likely to be skillful). We revised the text to clarify these points.

LL111-113: It is not particularly surprising that larger ENSO events lead to higher MHW forecast skill.

Agreed.

LL115-117: I don't understand this statement. The authors should clarify what they mean here.

Please see our response to previous comment about L105-106 and edits to this paragraph. The most relevant revised text is “Previous work has shown that ENSO is strongly tied to increased or decreased frequency of MHW occurrence in many regions¹², and while changes in the frequency of MHWs do not necessarily translate to changes in forecast skill (see methods and Extended Data Fig. 5), there is an ENSO-related modulation of MHW forecast skill.”

LL120-124: Do the authors really consider more skilful MHW forecasts would lead to a reduction in baleen whale entanglements? If so, how?

Of course, we don't know for sure, but there is reason to believe they could. One of the difficulties in managing fixed pot gear (in this case, for Dungeness crabs) is the delay in pulling it out of the water. A new study (Samhoury et al. 2021 Proc B; now added to references) highlights that even with real-time information on when risk to whales is greatest, the delayed response leads to both economic losses and high ecological risk. A forecast system could give both managers and fishers advance-warning so that gear could be removed before a high abundance of feeding whales moves inshore and converges with a high density of fixed gear pots.

LL124-127: The authors might also cite reference #11 (Holbrook et al. 2020) again here as that paper focuses on and provides a comprehensive summary of the benefits to fisheries and aquaculture managers of forecasts specifically from MHWs.

Added

LL135-149: This is interesting.

Thank you

LL147-149: Is removing the trend for a 3.5-month forecast really needed?

To clarify, here we are talking about the long-term warming trend, not the trend over the course of a 3.5-month forecast. We edited the sentence to clarify. It now reads:

“In this context, special consideration should be given to the handling of long-term SST trends in a forecast system, as the decision to retain or remove trends when defining MHWs will alter MHW frequency and consequently the statistics of forecast hits and misses (see methods and Extended Data Fig. 4).”

LL474-481 re Fig. 4: This is a nice concept figure. One thing though is that it mixes observed SSTa (Fig 4a) and 3.5-month MHW forecast probability thresholds (Fig. 4b), which are somewhat different metrics. Presumably the decision-making by the user should take account of forecast skill, not only forecast probability, which underpins management choices regarding risk tolerances to false negatives and false positives. An interesting concept.

Thank you! The observed SSTa are tied to the MHW forecast probability thresholds (we now clarify that the x-axes are the same for both panels). The top panel shows that a higher probability threshold isolates more intense MHWs, while the bottom panel shows how the probability threshold relates to forecast hits and misses.

LL491-495 re Extended Data Figure 2: In Extended Data Fig. 2a, it would be good to see forecasts for just south of Tasmania in the apparently predictable region outlined in Fig 1a.

We include here a panel for that region (shown for 147°E, 45°S), and we certainly appreciate your interest in the forecasts in your area. For clarity and consistency between figures we prefer not to add another panel in the manuscript – different individuals will likely want to explore

specific regions in more depth but here we've chosen a small subset to illustrate a range of MHW forecast characteristics.

LL496-500 re Extended Data Figure 3: I find it interesting that for the Tasman Sea region, according to this analysis, MHWs are poorly forecast through the summer (DJF) when they tend to be largest and most impactful.

Agreed, that is interesting, and it's an important point that the most intense MHWs are not necessarily the most predictable.

The reviewer is Neil Holbrook

Additional references

Brown, J.R., Lengaigne, M., Lintner, B.R., Widlansky, M.J., van der Wiel, K., Dutheil, C., Linsley, B.K., Matthews, A.J. and Renwick, J., 2020. South Pacific Convergence Zone dynamics, variability and impacts in a changing climate. *Nature Reviews Earth & Environment*, 1(10), pp.530-543.

Cai, W., Santoso, A., Wang, G., Yeh, S.W., An, S.I., Cobb, K.M., Collins, M., Guilyardi, E., Jin, F.F., Kug, J.S. and Lengaigne, M., 2015. ENSO and greenhouse warming. *Nature Climate Change*, 5(9), pp.849-859.

Callahan, C.W., Chen, C., Rugenstein, M., Bloch-Johnson, J., Yang, S. and Moyer, E.J., 2021. Robust decrease in El Niño/Southern Oscillation amplitude under long-term warming. *Nature Climate Change*, 11(9), pp.752-757.

Newman, M. and Sardeshmukh, P.D., 2017. Are we near the predictability limit of tropical Indo-Pacific sea surface temperatures?. *Geophysical Research Letters*, 44(16), pp.8520-8529.

Samhouri, J.F., Feist, B.E., Fisher, M.C., Liu, O., Woodman, S.M., Abrahms, B., Forney, K.A., Hazen, E.L., Lawson, D., Redfern, J. and Saez, L.E., 2021. Marine heatwave challenges

solutions to human–wildlife conflict. Proceedings of the Royal Society B, 288(1964), p.20211607.

Seager, R., Cane, M., Henderson, N., Lee, D.E., Abernathey, R. and Zhang, H., 2019. Strengthening tropical Pacific zonal sea surface temperature gradient consistent with rising greenhouse gases. Nature Climate Change, 9(7), pp.517-522.

Wengel, C., Lee, S.S., Stuecker, M.F., Timmermann, A., Chu, J.E. and Schloesser, F., 2021. Future high-resolution El Niño/Southern Oscillation dynamics. Nature Climate Change, 11(9), pp.758-765.

Reviewer Reports on the First Revision:

Referees' comments:

Referee #2 (Remarks to the Author):

I thank the authors for suitably addressing my previous concerns. In particular the demonstration of the sensitivities of prediction skill depending on whether MHWs are identified from daily or monthly data, illustrated for a selection of locations with different characteristics, seems acceptable.

It is however notable that the section that discusses these sensitivities has been written with a little less care than other parts of the manuscript, so that there remain a few places where clarification is needed (see specific comments below).

Also some methodological details are still unclear, and this currently hinders the reproducibility of the results. In particular, it should be clarified which sub-set of the 30-year hindcast period is used when calculating time series of the skill values (as in e.g. Extended Data Figure 5) - based on the understanding that skill values typically represent the skill calculated over all forecast initialization dates (such as in Fig. 1). Also it is unclear how exactly the warming trends have been removed in the analysis.

Specific comments:

- line 30: "reduced primary productivity" - could this be "reduced or increased primary productivity" (depending on region/event)?

- Methods line 60: the wording in the heading and the following section seems ambiguous, as the daily versus monthly scale should refer to the definition of MHW events, not the forecasts as the text seems to suggest. A more accurate heading could be e.g. "prediction sensitivity of identifying MHW from daily vs. monthly SST", or similar. Please make sure to also update the wording in the main text when referencing to this section

- Methods line 67: "...compare the daily and monthly forecasts" - in line with the previous comment, better say "forecasts of MHW identified based on daily and monthly data"?

- Methods line 71: "Lead-dependent" - should be "lead-time dependent" or similar?

- Methods line 75: unclear if you are referring to "daily/monthly forecasts", or "forecasts of events identified from daily/monthly data"?

- Methods line 80: I am not sure I understand what the authors mean here. It should be possible to predict (e.g. increased or reduced) probabilities of such shorter events at 5-day scale? It is true that deterministic predictions of individual events are typically not possible beyond the weather scale (approx 2 weeks). But the concept of seasonal to inter-annual climate predictions is to predict tendencies or probabilities for certain event types to occur.

- Methods line 81: "daily forecasts" or "events identified from daily data" or "events with duration of a few days"??

- Methods line 86: Readers may be confused about the statement that daily output from seasonal forecasts is generally not available, as C3S seems to provide SSTs from the seasonal predictions as 6-

hourly output: <https://cds.climate.copernicus.eu/cdsapp#!/dataset/seasonal-original-single-levels?tab=overview>

I understand that data availability may be a limitation of the models used in this study, but such general statement seems confusing.

- Methods line 100: How exactly has the detrending been done? Simple linear regression may not be suitable? Please clarify.

- Methods lines 108-181: It is currently not clear over which time steps (or using hindcasts from which initialisation dates/years) the skill metrics in the different figures were calculated. I understand that in seasonal predictions evaluation typically all available hindcasts are used, and the equations for BSS and FA explicitly note these measures were calculated over N forecasts (unclear: does this N refer to the number of available starting dates if hindcasts?). Based on this understanding, it is however unclear how the time series of the different skill measures (e.g. Extended Data Figure 5, or Figure 3b) were calculated. Is the skill value at each point in time calculated based on only a sub-set of the hindcasts (e.g. using a moving window around each date displayed in the quoted time series plots) - and if so, which ones?

- Extended data figure 5: how exactly has the trend been removed?

- Extended Data Figure 6: Based on the caption it is not clear what is really shown here. Is it daily/monthly forecast skill (as stated), or is it the skill in predicting MHWs defined based on daily/monthly SST data? Please also indicate the lead time for the skill measures shown here.

- Extended Data Figure 7: As above, is it daily/monthly forecast skill, or is it the skill in predicting MHWs defined/identified based on daily/monthly SST data?

- Extended Data line 58: "locations in Extended Data Fig. 7" - should this better refer to Extended Data Figure 6?

Referee #3 (Remarks to the Author):

I'm satisfied with the authors' point-by-point responses to my questions and comments, and manuscript revisions. This will make a valuable contribution to the literature.

Neil Holbrook.

Author Rebuttals to First Revision:

Referee #2 (Remarks to the Author):

I thank the authors for suitably addressing my previous concerns. In particular the demonstration of the sensitivities of prediction skill depending on whether MHWs are identified from daily or monthly data, illustrated for a selection of locations with different characteristics, seems acceptable.

It is however notable that the section that discusses these sensitivities has been written with a little less care than other parts of the manuscript, so that there remain a few places where clarification is needed (see specific comments below).

Also some methodological details are still unclear, and this currently hinders the reproducibility of the results. In particular, it should be clarified which sub-set of the 30-year hindcast period is used when calculating time series of the skill values (as in e.g. Extended Data Figure 5) - based on the understanding that skill values typically represent the skill calculated over all forecast initialization dates (such as in Fig. 1). Also it is unclear how exactly the warming trends have been removed in the analysis.

Thank you for the additional feedback on our manuscript. We are pleased that your major concerns have been addressed, and we have clarified these remaining points as detailed under specific comments below.

Specific comments:

line 30: “reduced primary productivity” - could this be “reduced or increased primary productivity” (depending on region/event)?

We changed the wording to “altered primary productivity” to allow for both directions of change.

Methods line 60: the wording in the heading and the following section seems ambiguous, as the daily versus monthly scale should refer to the definition of MHW events, not the forecasts as the text seems to suggest. A more accurate heading could be e.g. “prediction sensitivity of identifying MHW from daily vs. monthly SST”, or similar. Please make sure to also update the wording in the main text when referencing to this section

Good point. We renamed this section “Sensitivity to defining MHWs from daily vs. monthly SST”, and updated the main text as well as other sections of the methods (as noted under comments below).

Methods line 67: “...compare the daily and monthly forecasts” - in line with the previous comment, better say “forecasts of MHW identified based on daily and monthly data”?

Changed

Methods line 71: “Lead-dependent” - should be “lead-time dependent” or similar?

Methods line 75: unclear if you are referring to “daily/monthly forecasts”, or “forecasts of events identified from daily/monthly data”?

We deleted “lead-dependent” from this sentence and clarified the language about daily/monthly forecasts: “Skill metrics for MHW forecasts generated from daily SST output were calculated using the same methods as those applied to monthly SST forecasts (see “Marine Heatwave Forecast Evaluation” below).”

Methods line 80: I am not sure I understand what the authors mean here. It should be possible to predict (e.g. increased or reduced) probabilities of such shorter events at 5-day scale? It is true that deterministic predictions of individual events are typically not possible beyond the weather scale (approx 2 weeks). But the concept of seasonal to inter-annual climate predictions is to predict tendencies or probabilities for certain event types to occur.

Agreed, the forecasts can predict increased/decreased tendency for shorter-lived events to occur. The point we were trying to make is that one cannot predict the detailed evolution of a specific short-lived event that far in advance. We now make that point more explicitly: “it is important to note that even though seasonal forecasts can predict enhanced or reduced likelihood of MHWs on daily timescales, this does not mean that one can predict the details of a specific short (e.g., 5-day) warming event months in advance.”

Methods line 81: “daily forecasts” or “events identified from daily data” or “events with duration of a few days”??

Changed to “MHW forecasts provided at daily resolution”

Methods line 86: Readers may be confused about the statement that daily output from seasonal forecasts is generally not available, as C3S seems to provide SSTs from the seasonal predictions as 6-hourly output: <https://cds.climate.copernicus.eu/cdsapp#!/dataset/seasonal-original-single-levels?tab=overview>

I understand that data availability may be a limitation of the models used in this study, but such general statement seems confusing.

Thanks for pointing this out. We revised the statement to be more precise: “We also note that daily output is often not publicly available for seasonal forecasts (e.g., NMME output is provided as monthly averages).”

Methods line 100: How exactly has the detrending been done? Simple linear regression may not be suitable? Please clarify.

We now specify that the detrending is done by removing a linear trend from the observed and forecast SST anomalies at each grid cell. It is true that over longer records, especially in future temperature projections when SST warming accelerates, something like quadratic detrending may be more suitable. But for our 30-year historical analysis based on observations, a linear trend is appropriate. In this relatively short historical record, forced trends are near constant,

and natural variability can drive short term trends that could be incorrectly identified as nonlinear secular change (e.g., DelSole et al. (2011), “A significant component of unforced multidecadal variability in the recent acceleration of global warming”, J. Climate, 24(3), 909-926).

Methods lines 108-181: It is currently not clear over which time steps (or using hindcasts from which initialisation dates/years) the skill metrics in the different figures were calculated. I understand that in seasonal predictions evaluation typically all available hindcasts are used, and the equations for BSS and FA explicitly note these measures were calculated over N forecasts (unclear: does this N refer to the number of available starting dates if hindcasts?). Based on this understanding, it is however unclear how the time series of the different skill measures (e.g. Extended Data Figure 5, or Figure 3b) were calculated. Is the skill value at each point in time calculated based on only a sub-set of the hindcasts (e.g. using a moving window around each date displayed in the quoted time series plots) - and if so, which ones?

Your understanding for the first part is correct – skill metrics are calculated over all available start dates. In that case, we are looking at skill over time for each specific location.

For the time series, we include new text in the “Marine heatwave forecast evaluation” section to clarify the method: “When calculating time series of forecast skill (Fig. 3b, Extended Data Fig. 5), skill metrics are calculated over all grid cells at each time, rather than over all times at each grid cell. For example, the forecast accuracy for a given month in Fig. 5b is the fraction of the ice-free global ocean for which the MHW state that month was correctly predicted.”

Extended data figure 5: how exactly has the trend been removed?

We revised the caption to clarify that a linear 1991-2020 trend was removed.

Extended Data Figure 6: Based on the caption it is not clear what is really shown here. Is it daily/monthly forecast skill (as stated), or is it the skill in predicting MHWs defined based on daily/monthly SST data? Please also indicate the lead time for the skill measures shown here. Extended Data Figure 7: As above, is it daily/monthly forecast skill, or is it the skill in predicting MHWs defined/identified based on daily/monthly SST data?

In both captions, we clarified that we are referring to the daily/monthly resolution of SST data. For extended data figure 6, the lead time is 3.5 months, which is now included in the caption.

Extended Data line 58: “locations in Extended Data Fig. 7” - should this better refer to Extended Data Figure 6?

Yes, thank you for catching the error!

Reviewer Reports on the Second Revision:

Referees' comments:

Referee #2 (Remarks to the Author):

I thank the authors for clarifying these remaining points, and am happy to recommend this manuscript for publication.

One last comment regarding the use of linear detrending in this study. Previous studies have documented issues with this method introducing biases in time series affected by internal variability and external forcing (see e.g. Frankcombe et al 2015; <https://doi.org/10.1175/JCLI-D-15-0069.1>). While I do not believe that this methodological choice would substantially alter the results of this study, I at least want to raise this to the authors for their consideration as a potential criticism by readers. I leave it to the authors whether and how they may want to address this issue.